# Pluralistic Seed System Development: A Path to Seed Security?

**Teshome Hunduma Mulesa** [1,*] , **Sarah Paule Dalle** [2] , **Clifton Makate** [3] , **Ruth Haug** [1] and **Ola Tveitereid Westengen** [1]

1   Department of International Environment and Development Studies, Noragric, Norwegian University of Life Sciences, Box 5003, 1432 Ås, Norway; ruth.haug@nmbu.no (R.H.); ola.westengen@nmbu.no (O.T.W.)
2   Development Fund of Norway, Mariboes Gate 8, 0183 Oslo, Norway; sarah@utviklingsfondet.no
3   School of Economics and Business, Norwegian University of Life Sciences, Box 5003, 1432 Ås, Norway; clifton.makate@nmbu.no
*   Correspondence: teshome.mulesa@nmbu.no

**Abstract:** Seed security is central to crop production for smallholder farmers in developing countries, but it remains understudied in relation to long-term seed sector development. Here, we compare seed systems in two districts of Central Ethiopia characterized by subsistence-oriented teff cultivation and commercially oriented wheat production and relate this to the country's pluralistic seed system development strategy (PSSDS). Our analysis is based on quantitative and qualitative information from a household survey and focus group discussions with farmers, as well as document review and key informant interviews with actors that make up the seed sector in the study sites. Farmers in both districts used a range of seed sources but primarily obtained their seeds from informal sources. Evidence of seed insecurity was found in both districts, as apparent from discrepancies between what the seed farmers say they prefer and those they actually use, limited availability of improved varieties and especially certified seeds of these, challenges with seed quality from some sources, and differentiated access to preferred seed and information according to sex, age and wealth. We find that the interventions prioritized in the PSSDS address most of the seed security challenges and seed system dysfunctions identified, but implementation lags, particularly for the informal seed system, which is largely neglected by government programs. The intermediate system shows promise, but while some improvements have been made in the formal system, vested political, organizational, and economic interests within key institutions represent major obstacles that must be overcome to achieve truly integrative and inclusive seed sector development.

**Keywords:** seed security; access to seeds; seed availability; seed quality; varietal suitability; seed policy; smallholder farmers; Ethiopia

## 1. Introduction

Access to good seeds is fundamental for smallholder farmers' crop production and resilience in the face of environmental change and disasters. National seed policies and programs in developing countries have predominantly focused on the formal seed supply system, but despite decades of efforts to spur a Green Revolution in Sub Saharan Africa, farmers' use of seeds from the formal seed system remains limited [1]. In 2017, Ethiopia was the first country to officially adopt a Pluralistic Seed System Development Strategy (PSSDS) as an alternative to the dominant linear approach, i.e., formal seed system development [2,3]. The strategy is pluralistic in that it proposes support for three major seed systems operating in the country (informal, formal and intermediate) and promotes complementarity between value-chain components of each seed system. In this study of the Ethiopian seed sector, we analyze farmers' seed security and discuss the relevance and implementation of the new policy in terms of addressing farmers' challenges with access to enough quality seeds of preferred crops and varieties.

Farmers' access to seed is increasingly theorized in terms of two closely related concepts: seed systems and seed security [4,5]. The seed system concept has deep roots,

and various fields from crop science to agricultural anthropology and economics have contributed to the current understanding of seed systems as the activities, institutions, and actors involved in the development, distribution, and use of seeds [6–12]. This literature has highlighted that farmers in developing countries source most of their seeds outside the formal system, which develops and approves improved varieties, and regulates seed quality assurance and certification. Consequently, a branch of this literature suggests that efforts to support farmers' access to seeds should recognize the complementarity of formal and informal seed systems and thus advocates a pluralistic approach to seed sector development by promoting complementarity of activities between value-chain components of each seed system [5,13–17]. The Ethiopian PSSDS is arguably among the first national seed policies to take this perspective on board.

*Seed security* is a more recent concept originating in the literature on emergency seed aid in the wake of natural and human-made disasters [4,18–20]. The Food and Agriculture Organization of the United Nations (FAO) defines seed security as "ready access by rural households, particularly farmers and farming communities, to adequate quantities of quality seeds and planting materials of crop varieties, adapted to their agroecological conditions and socioeconomic needs, at planting time, under normal and abnormal weather conditions" [21]. The conceptual frameworks for seed security initially were based on three basic dimensions: *availability, access, and quality* (including seed quality and variety quality) [22]. Recently, FAO has added the two dimensions *varietal suitability* (varietal traits responding to farmers' preferences, previously included under the "quality" parameter) and *resilience* (stability of seed system in the context of shocks) to their framework [19,23]. Several frameworks and tools have been developed by researchers and practitioners concerned with understanding barriers and options for strengthening farmers' seed security [19,23–26]. The application of such frameworks has arguably led to relief efforts better tailored to specific local contexts [27]. For research more generally, the seed security concept (and related frameworks) provides a lens through which the performance of each seed system can be assessed. In this sense, seed security can be understood as a livelihood concept, representing the outcome of seed systems from the farmers' perspective [20,28]. Analysis of the roles and interactions between different actors in the seed sector is key to understanding seed security [27]. However, few studies analyze the complex interplay of policy, institutional, socio-economic, technical, and household-level factors that underly seed security challenges. Research linking the performance of seed systems to seed security outcomes, while considering the range of seed systems and channels farmers use, is therefore needed to deepen our understanding of the context-specific conditions and vulnerabilities that affect seed security, as well as to inform policy formulation [29].

Post-disaster seed security studies have shown that pre-existing "chronic stresses" are often at the root of most seed security problems [30]. While, in theory, the seed security concept is as applicable to understand the performance of seed systems in both normal seasons and those affected by disasters [23,25], there are few examples of seed security assessments from non-emergency contexts [20]. Studies analyzing the functioning of seed systems in developing countries under normal conditions [6,31–34] rarely empirically assess their effect on seed security [35]. Rather, most of the research on seed use in non-emergency contexts has solely focused on barriers to and determinants of adoption of improved varieties from the formal system [36–42]. This econometric literature commonly shows that women are less likely to adopt improved varieties than men [37] due to lack of access to key resources such as land, cash, credit, labor, and extension services [43,44] as well as challenges related to gender roles within households and communities [44,45]. Furthermore, a common finding is that the likelihood of adopting improved varieties increases with wealth [38,46,47], while the effect of age varies [48]. This adoption literature provides valuable assessments of supply and demand in the formal seed system, but its perspective does not suffice for assessing factors influencing seed use outside the formal system. From a seed security perspective this is a major gap as the formal system only covers a small share of farmers' seed use. In this article, we aim to address this gap by

exploring the relationship between farmers' seed security and the functioning of the seed systems they use under normal conditions in the central highlands of Ethiopia.

Ethiopia is a crop diversity hotspot, and a large body of literature exists both on the nature and geography of this diversity and the seed systems farmers use [32,34,49–51]. A few seed security assessments have been conducted to guide seed-related interventions [52,53], but the academic literature has made limited use of the seed security framework to analyze Ethiopian seed systems. The importance of crop diversity and local seed system is recognized in Ethiopia's national policy and law [54–56], and, as stated above, in 2017, Ethiopia became the first country to officially adopt a pluralistic seed system development strategy (PSSDS) as policy. Ethiopia's unique PSSDS, with provisions to support both formal and informal, as well as an emerging "intermediate" seed system, makes it a very interesting case to examine how the different seed systems function and their impacts on farmers' seed security.

In this context, we analyze farmers' seed use and preferences (demand-side) and the role of supply side institutions and actors, to understand how different elements of the seed systems affect farmers' seed security (i.e., varietal preferences, seed quality, and the availability and access of seeds from different sources). Specifically, we address the following research questions: (1) How does farmers' seed security differ between commercially and subsistence-oriented production systems; (2) How do wealth status, gender, and age affect farmers' access to preferred seeds from different seed systems; and (3) To what extent does Ethiopia's pluralistic approach hold potential to improve farmers' seed security and how is this conditioned by institutional, political and economic interests?

We address these questions using a comparative case study of two districts in the central highlands of Ethiopia with similar agroecological contexts but contrasting degree of seed system formalization and commercialization. The selected districts represent the range of conditions that smallholder farmers in Ethiopia face and provide a good basis for understanding how different elements of the informal, formal, and intermediate seed systems impact seed security.

The paper is organized as follows. First, we provide an overview of Ethiopia's PSSDS, as well as our methodology, study sites, and the crops and seed sector actors engaged in each district. We then present a comparative analysis of the dimensions of seed security in the two districts as experienced by smallholder farmers on the ground, considering household differences in access to preferred seeds. Thereafter, we map key seed sector actors in the study areas and analyze their roles and performances in seed supply and seed system governance to understand to what extent the priorities set out in Ethiopia's PSSDS address the seed security challenges identified in the previous section. In addition, we analyze the political, organizational, and economic factors that affect the implementation of the PSSDS, as revealed by our empirical findings on the performance of different actors. To conclude, we draw key lessons from this study on what it takes to achieve a pluralistic seed system development.

## 2. Ethiopia's Pluralistic Seed System Development Strategy

For decades, the Ethiopian government followed a linear model of formal seed sector development policy focusing on the development of improved high-yielding varieties and the distribution of certified seeds to farmers to increase national food security [35,57–61]. This approach started to be questioned in policy debates in the 1990s [62,63]. By the mid-2000s, the government policy began to shift, leading to the development of the first version of the PSSDS in 2013 [3]. This process was supported by the Integrated Seed System Development program (ISSD), initiated in Ethiopia in 2009, and informed by critical evaluations of the country's policies and programs [33,64] and experiences from community-based seed production projects within Ethiopia [65–70]. The ISSD program is part of the "Bilateral Ethiopian Netherlands Effort for Food, Income and Trade Partnership (BENEFIT Partnership) supported by the Dutch Government through the Embassy of the Kingdom of the Netherlands in Addis Ababa since 2009. The program is operationalized

by the Centre for Development Innovation of Wageningen University & Research Centre and the Royal Tropical Institute (KIT), the Netherlands. It is implemented in the context of the African Seed and Biotechnology Programme of the African Union Commission (African Union 2008) through its local partners in Ethiopia, Mozambique, Nigeria and Uganda. With Ethiopia's PSSDS, the previous policy focus of replacing the informal seed system with the formal seed system changed to supporting the diverse seed systems farmers use, exploiting both market and non-market channels for increasing seed security. This includes policy recognition of the existence of three different seed systems—informal, formal, and intermediate—which all have different performances in terms of seed security for different crops [2,3].

The informal seed system involves farmers' seed selection, multiplication, storage, use, and distribution through social seed networks and local markets. It dominates in terms of delivering large quantities of seeds of a diversity of crop varieties [28,31,59,71,72]. This includes both traditional varieties and improved varieties that have been released by the formal system in the past and integrated into the local seed system, so-called "obsolete" improved varieties [32]. The formal seed system involves public and private sector institutions and a linear series of activities along the seed value chain, including germplasm conservation in genebanks, plant variety development, variety release and registration, quality seed production, and distribution [58]. It plays a crucial role in delivering certified seeds of improved varieties of certain crops, including maize and wheat [73–75]. The formal system is still at an early stage of growth and is dominated by public institutions [1]. Additionally, an emerging intermediate seed system is growing in Ethiopia. This system involves market-oriented farmer groups who produce and market non-certified seeds of both improved varieties and farmer-preferred local varieties [65,76–78]. These community-based seed groups include Local Seed Businesses or Seed Producer Cooperatives (SPC) who produce quality declared seeds (QDS) of improved varieties. QDS is a simplified certification scheme developed by FAO in which seed-producing farmers are responsible for seed quality, while the government plays a monitoring role [79]. In Ethiopia, the QDS scheme requires seed producers to employ robust internal quality assurance and declare the quality of their seed based on limited quality control established by the regulatory authorities (Regional Bureaus of Agriculture), e.g., inspection of 10% of the total seed produced instead of undergoing the full inspection and quality testing procedures. This has intended to reduce the burden on seed regulatory authorities and hasten community-based production and marketing [55]. In addition, the intermediate seed system includes non-profit community-based seed producers such as community seed bank (CSB) groups [80] who produce higher quality seed than typically produced by the informal system, even if it is not certified nor fully regulated under existing regulations [3].

The PSSDS was fully adopted by the Ministry of Agriculture in 2017 [2], and based on this strategy, the government subsequently revised the national seed policy [54]. The government has also developed/amended a series of laws and regulations [2] including: (1) A Plant Variety Protection or a Plant Breeders Rights law to encourage the development of commercial plant varieties [56]; (2) A national seed law and regulation for commercial seed production and distribution of certified seeds [81,82]; (3) A QDS scheme and community based seed (CBS) production directive for multiplication and distribution of non-certified seeds of either improved or local varieties within the local community or nearby communities [55]; and (4) several other service and governance related directives concerning seed marketing. These service and governance related directives include the Council of Ministers Regulation to Determine the Rate of Fee for Seed Competency and Related Services Proclamation No. 361/2015, the Directive for Issuance and Administration of Certificate of Competency Proclamation No. 02/2010 and the Directive for tracking rejected seed field and lot Proclamation No. 03/2010.. The informal seed system is left unregulated, but interventions were identified to strengthen the system, emphasizing on the key seed security features [2,3]. We return to the PSSDS provisions in Section 6 of this paper when we discuss its match with farmers' seed security needs.

## 3. Methods

This study is based on fieldwork conducted from October 2017 to February 2018 in a total of eight gandas in Gindabarat and Heexosa districts (four gandas per district). *Ganda* is the smallest administrative unit in Oromia National Regional State of the Federal Democratic Republic of Ethiopia. This administrative unit is called "kebele" in other parts of the country. Methods included a household survey and focus group discussions (FGDs) with small-holder farmers, key informant interviews with seed sector actors in the respective gandas/districts, and document analysis.

In order to assess actors' roles and performances in seed supply and seed system governance, we used the CGIAR Roots, Tubers, and Bananas program's "multi-stakeholder framework intervention in RTB seed systems" [26]. This is an actor-oriented approach, in which the roles of seed sector actors are analyzed in relation to different seed security parameters.

For this study, the analysis focused on the following actors: local government and extension services, regulatory bodies/seed laboratories, national/regional agricultural research, international research, local traders, public seed enterprises, agrodealers, SPCs and farmers' unions, Non-governmental organization (NGOs) and development agencies, private sector grain processors and smallholder farmers. Information on seed supply and seed system governance was collected from these actors using FGDs with 80 smallholder farmers (see details below) and semi-structured interviews with 50 key informants. A checklist for the FGDs and key informant interviews was developed covering the following topics: seed use and management, seed availability, access, quality, and varietal suitability, farmers' resilience to shocks, technological and institutional innovation, historical policy and institutional changes, and actors perceptions and roles in the seed sector. Questions were tailored for specific actors and elicited information on both the current situation and changes over time, where appropriate. All FGDs and key informant interviews were recorded, transcribed, and analyzed using the RTB matrix (Table A1).

The demand side of farmers' seed security was assessed using quantitative data from the household survey, complemented with qualitative information from the FGDs. The household survey was administered to a stratified random sample of 223 household heads in Gindabarat and 209 in Heexosa. The sampling frame was established from a list of household heads, and stratified by wealth category (poor, medium, rich), age and gender, based on information provided by the ganda administration. Households were then randomly selected from each stratum. In cases where the randomly selected household was not available, another household from the same stratum was interviewed. Focusing on the 2017/2018 main growing season (June to September), the survey elicited quantitative information on the types of seeds and seed sources farmers used. It also produced quantitative information on household characteristics, agricultural assets, labor, and other biophysical factors. The survey instrument drew on tools developed for seed security assessment [19,23,25] and for seed sector and seed value chain analysis [83,84] to assess varietal suitability, seed availability, seed access, and seed quality. Statistical analysis was conducted using STATA version 15 [85].

The FGDs were conducted with men and women household heads in all eight survey gandas (16 FGDs). Participants were purposively selected from the stratified random sample used for the household survey. Separate FGDs were held with women and men, with representation from all wealth and age groups. In the case of female heads of household (FHH), these were mainly widows and divorcees, a few of whom were women in polygamous relationships who essentially functioned as FHHs. In total, over 80 farmers contributed to the qualitative empirical data in this study.

## 4. Study Area, Crops, and Actors

The study was conducted in Heexosa district in Arsi Zone and Gindabarat district in West Shewa Zone of Ethiopia's Oromia Regional State (Figure 1, Table 1).

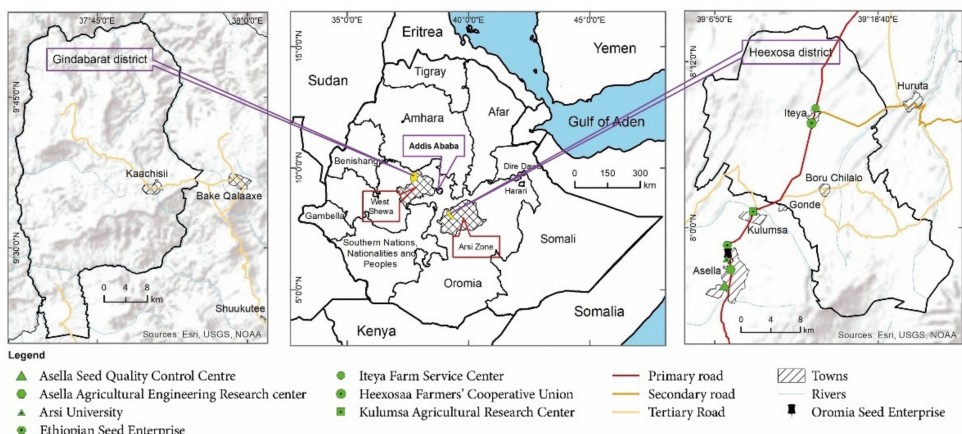

**Figure 1.** Map of Ethiopia, Gindabarat and Heexosa districts.

The study districts were selected to represent contrasting degrees of seed system formalization and commercialization, considering both institutional and physical infrastructure (Figure 1), but with otherwise similar conditions in terms of landholding size, agroecological and demographic characteristics (Table 1). Gindabarat is remote, being isolated geographically by lowland gorges and rivers which separate it from all but one neighboring district, and physically, due to a poor road network. Gindabarat lacks research and proper institutional services that facilitate access to agricultural technologies, including fertilizer and improved seeds. Heexosa, on the other hand, is centrally located in terms of access to primary and feeder roads and linkages with institutions providing inputs, credit, and marketing services. In the late 1960s, Arsi zone was selected as one of the areas in Southeastern Ethiopia for the first green revolution project that focused on bread wheat cultivation, and already by 1972, about 150 landowners in Arsi were operating more than 250 tractors and 50 combines on approximately 30,000 hectares of land [86]. Nowadays, 97% of farmers in Heexosa use combine harvesters, as opposed to threshing their wheat crop manually [87].

The difference in formal seed system development in the two districts is reflected in what crops farmers cultivate. The Ethiopian staple grain teff (*Eragrostis tef*) is the key crop in Gindabarat, while in Heexosa, nearly all farmers produce bread wheat (*Triticum aestivum*) (hereafter wheat). In both districts, FGD participants identified a high infra-specific diversity of the dominant crop by their vernacular/cultivar/breed names (27 teff varieties in Gindabarat and 25 wheat varieties in Heexosa), with individual households growing on average three to four varieties of their key crop in the 2017/18 growing season.

In Gindabarat, farmers mainly planted local varieties of teff (68% of seed sown), although one improved variety of teff (Quncho) is popular. For wheat, old improved varieties that have been integrated into the local seed system (obsolete varieties) were the dominant (57% of seeds), while the remaining varieties are recycled seeds of improved varieties recently supplied through the Primary Multipurpose Cooperatives (PMCs) in Gindabarat. In Heexosa, farmers relied primarily on improved varieties of both wheat and teff (89% and 64% of seed, respectively) (Figure 2). "Local" wheat varieties in Heexosa are mostly obsolete improved varieties that were recycled for more than five years. In order to distinguish between obsolete and improved varieties, we used a five-year cut-off point based on recommendation from wheat breeders at the International Maize and Wheat Improvement Center (CIMMYT) in Addis Ababa and Kulumsa Agricultural Research Center. Thus, we considered improved varieties to be those that farmers recycle up to five years, while local varieties were improved seeds recycled for more than five years and traditional varieties.

**Table 1.** Key demographic and agroecological characteristics of Gindabarat and Heexosa districts.

| Characteristics | Districts | |
|---|---|---|
| | Gindabarat | Heexosa |
| Total population | 104,595 [a] | 124,219 [a] |
| Population (persons/per km)[2] | 124 [a] | 188 [a] |
| Rural Population | 90% [a] | 85% [a] |
| Total land/Crop land (ha) | 119,879/65,491 [b] | 93, 700/49,498 [c] |
| Major crops cereal and pulse crops in order of total production | Teff, maize, sorghum, wheat, faba bean, barley and field peas [d] | Wheat, barley, maize, faba bean, teff, sorghum and field peas [d] |
| Elevation (masl) | 1501–2607 [e] | 1500–4170 [f] |
| Topography | Plateau, hilly and sometimes steep slopes [e] | Mostly flat terrain [f] |
| Climate | Highland (temperate) and midland (moist sub-tropical) accounting for 40% and 60% of the area, respectively [e] | Highland (temperate), midland (moist sub-tropical) and midland (dry sub-tropical) accounting for 17%, 61% and 22% of the area, respectively [f] |
| Mean maximum and minimum annual temperatures (°C) | 10–25 [e] | 14–27 [h] |
| Mean farm size (ha) | 2.15 [g] | 2.31 [h] |
| Households with 0/1/2/ > 2 oxen (%) | 7/6/49/37, respectively [h] | 8/27/44/21, respectively [h] |
| Annual minimum and maximum rainfall at district town (mm) | 1377.9 to 2214.2 [i] | 800–1300 [f] |
| Rainfall onset | Low variability with 12.1% coefficient of variation. Receive most rainfall during long rainy season (June to September) [i] | Low variability except in dry mid-land areas. Receive most rainfall during long rainy season (June to September) and some during short rainy season (February to May) [f] |

[a] CSA [88]; [b] Amenu et al. [89]; [c] Yiemene [90]; [d] CSA [91] [e] Mulesa and Mulubiran [92,93]; [f] Amade and BFED [94,95]; [g] Beressa [96]; [h] Respective District agriculture bureaus; [i] Nurgi [97].

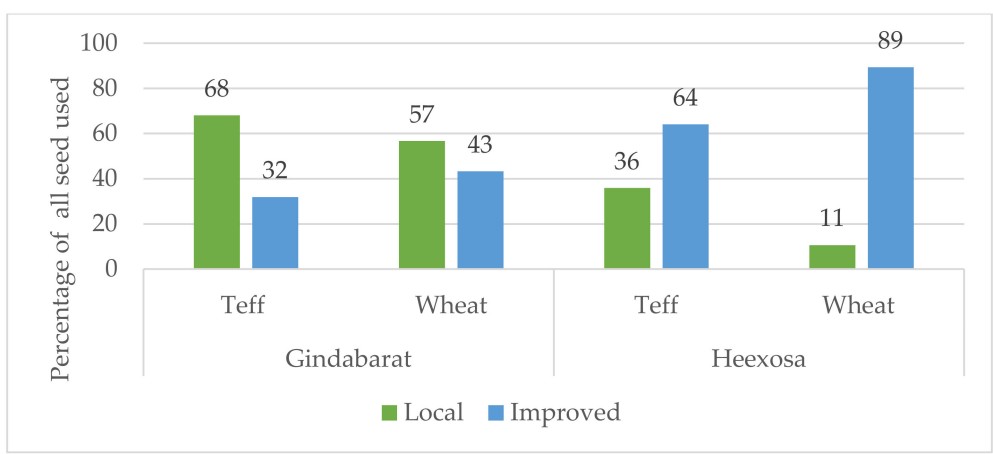

**Figure 2.** The use of teff and wheat varieties by farmers (% of all seeds used) in Gindabarat (*n* = 222 teff growers sowing 11,428.00 kgs of seeds on 297.36 hectares of land and *n* = 28 wheat growers sowing 676 kgs of seeds on 6.03 hectares of land) and Heexosa (*n* = 207 wheat growers sowing 85,149.00 kgs of seeds on 342.15 hectares of land and *n* = 60 teff growers sowing 1756.00 kgs of seeds on 27.64 hectares of land) districts during 2017/2018 growing season. Improved seeds category includes certified seeds and recycled seeds up to five seasons.

A range of institutions are involved in the development, production, and dissemination of seeds in the two areas. There are some significant differences between the two

districts, both in terms of the actors present and their level of engagement, reflecting the different degrees of formalization and commercialization of the seed sector (Table 2). In both districts, farmers are the backbone of the seed sector, with the district bureaus of agriculture, traders, NGOs, and CSB groups having lesser but similar levels of engagement. Many actors engaged in the formal seed system are only present in Heexosa. These include agriculture research, commercial seed producers, regulatory bodies, processors, and distributors for quality declared and certified seeds. National agriculture research centers, farmers' unions, and their member PMCs play a much more important role in Heexosa than in Gindabarat (Table 2, Table A1).

**Table 2.** Actors engaged in seed supply and seed sector governance in the study districts. Our assessment of the actors' contribution to smallholder farmers' seed security is indicated as high (***), moderate (**) or low (*). Actors that are not operating in the districts or are not engaged in seed supply and seed sector governance are denoted with (–). See Aix A (Table A1) for details.

| Actors | Gindabarat | Heexosa |
|---|---|---|
| 1. Smallholder farmers/households | *** | *** |
| 2. National agricultural research centers | * | *** |
| 3. International research centers (e.g., CIMMYT) | – | *** |
| 4. Seed producer cooperatives | – | *** |
| 5. Regional agricultural research institutes | – | ** |
| 6. Regulatory bodies/seed quality control and certification laboratories | – | ** |
| 7. Agro-dealers/retail sales outlets | – | * |
| 8. Private sector grain processors | – | * |
| 9. Commercial private farms | – | * |
| 10. Public seed enterprises | – | ** |
| 11. District bureau of agriculture | ** | ** |
| 12. Grain/seed traders (include farmers) | ** | ** |
| 13. Farmers' Union and primary multi-purpose cooperatives | * | ** |
| 14. Non-Governmental Organizations | * | ** |
| 15. Community seed bank groups | * | * |
| 16. Afoosha ‡ | * | – |

‡ Afoosha is an indigenous social institution established to provide financial and other types of support when a family member dies in most communities in Ethiopia. In Gindabarat, we found that Afoosha groups have established grain reserve in most peasant associations to support poor families affected by calamities by providing food grains, which is increasingly used by those affected as seeds

## 5. Assessing Demand-Side Seed Security

### 5.1. Varietal Suitability

Varietal suitability refers to whether crop varieties have traits that meet farmers' specific needs and preferences, such as yield, storability, marketability, tolerance to environmental stresses, pests and diseases, and culinary and cultural needs [19,29,32,98]. In terms of seed security, problems of varietal suitability are generally associated with chronic conditions, such as the buildup of pests and diseases, genetic erosion, and lack of access to extension/research services [99–101]. In addition, the distribution of varieties that are poorly adapted or fail to meet farmers' preferences is a common problem in seed relief and agricultural extension efforts [11,15,30,102].

To understand farmers' varietal preferences, we asked survey respondents to list all varieties of their key crop they grew and rate each according to a set of criteria. The criteria were: agroecological adaptation (tolerance to drought and frost, and resistance to plant diseases), socio-economic importance (household food security, yield, fodder value,

grain market value and cost of agrochemical inputs), and culinary and cultural uses (taste). This was triangulated with qualitative information on varietal preferences collected in the FGDs, which in all cases was found to be consistent. In both districts, respondents preferred at least one improved variety of their key crop, but the overall importance of improved compared to local varieties was higher in Heexosa (Table 3). In Gindabarat, 42% of respondents preferred the improved variety Quncho, released in 2006, but the remaining preferred teff varieties were all local. In contrast, most of the wheat varieties preferred by respondents in Heexosa were improved varieties released during the past decade, except Kubsa, which was released in 1995.

**Table 3.** Widely grown and preferred varieties of teff by proportion of respondents in Gindabarat (*n* = 222) and Heexosa (*n* = 207) and by area coverage.

| | Variety Name (Year Released) | Variety Type | Proportion of Respondents | Total Area Sown (ha) |
|---|---|---|---|---|
| Teff in Gindabarat | Quncho (2006) | Improved | 42% | 81.8 |
| | Daaboo | Local | 30% | 29.9 |
| | Adii-qola-gurraachaa | Local | 22% | 47.2 |
| | Adii-qola-adii | Local | 22% | 52.6 |
| | Minaaree | Local | 13% | 22.1 |
| | Maanyaa | Local | 11% | 21.0 |
| Wheat in Heexosa | Ogolcho (2012) | Improved | 59% | 125.4 |
| | Kubsa (1995) | Improved | 55% | 91.5 |
| | Hidase (2012) | Improved | 52% | 75.5 |
| | Kingbird (2015) | Improved | 18% | 31.4 |
| | Kakaba (2010) | Improved | 13% | 18.9 |

Farmers' varietal preferences were shaped by a combination of agroecological, socio-economic, and cultural factors. For example, in Gindabarat, Maanyaa and Quncho are both white-seeded varieties that fetch a high market price due to urban consumers' preference for lighter *buddeena*. *Buddeena* (Oromo) or *enjera* (Amharic) is a fermented flat bread that is a staple food in many parts of Ethiopia. Quncho is high yielding with good straw palatability for cattle and equines but is only adapted to midland agroecology. FGD participants explained that Quncho has good vegetative growth in the highlands at the expense of seed-bearing panicles and fails to yield enough grain/seed. In contrast, Maanya is low yielding but is widely adapted. Daaboo is a brown-seeded variety with lower market value but is well adapted to both highland and midland agroecological areas of Gindabarat. According to FGD participants, Daaboo is higher-yielding than all white-seeded varieties and has good taste and nutritional quality, as expressed by the following local proverb in the Oromo language: "*Daaboo dhiiga dhiiraa, dhiirrii qoomaf, dubartiin duugdaf si nyaattii*", meaning "Daaboo, you are part of men's blood, men eat you for physical strength; women eat you to regain back strength [after labor/delivery]".

When explaining the challenges they faced in terms of varietal suitability, FGD participants in Gindabarat mentioned the loss of local varieties due to their susceptibility to new plant diseases (e.g., wheat rust) and climatic variability (e.g., late onset of rain) as well as the absence of new, improved varieties that are adapted to these challenges. Aside from Quncho and the obsolete wheat varieties, improved varieties are totally lacking in the district. FGD participants described the chronic varietal insecurity in several food crops as follows:

> *In the past, we had many traditional varieties of teff, wheat, barley, sorghum, maize, peas, and faba bean. Most people have now abandoned many traditional varieties, especially*

*sorghum and wheat. Unfortunately, we do not get disease-resistant or well-adapted improved seeds from the government. So, we shifted to teff and maize. We also have a bad experience with the few varieties of teff and sorghum that we received from the agriculture bureau in the past. Almost all failed to perform well on our soil. A few years ago, a new sorghum variety did not flower at all. It failed. We are now cautious about using new varieties because the risk is high if it fails after investing all our resources (labor, seeds, fertilizer, and land) into its cultivation. The two most important improved varieties that have benefited us so far are Quncho and hybrid maize varieties.*

Elderly FGD participants stated that chronic varietal insecurity in wheat represented a huge production loss for farmers in Gindabarat compared to three to four decades ago when wheat was widely grown. Even as recently as 2006, the proportion of households growing wheat and number of wheat varieties was much higher than at the time of the present study (31% vs. 13% households and 14 vs. 6 varieties) [92].

In Heexosa, farmers generally preferred improved wheat varieties released in the last decade due to their yield and relative wheat rust resistance, although many respondents also selected wheat varieties based on other factors such as market value, taste, frost tolerance, and straw palatability for livestock. The most striking example is Kubsa, which continues to be planted in Heexosa for its high yield and good taste, despite being susceptible to wheat rust and requiring frequent application of pesticides. That said, FGD participants explained that the virulence of the Ug99 stem rust was a major concern and strongly emphasized the need for continuous varietal replacement:

*Our biggest concern is the recent increase in wheat rust [i.e., Ug99]. There were plant diseases in the past too. Now it is worse. We see a link between climatic variation, such as the late onset of rain, and wheat rust. When we observe rust on maize in June following a late rain, we know that it will be devastating for wheat in the autumn. In the past 10 years, if it had not been for pesticide, we would not have produced even for our own consumption. Thanks to pesticides, we now produce a surplus for the market. The day Kulumsa Agricultural Research Center is unable to develop rust-resistant varieties for us, and agrodealers stop the supply of pesticides, agriculture will collapse in our district. We cannot go back to traditional varieties for better resistance and higher yield. Most traditional wheat varieties lodge if we apply fertilizer because they grow tall and have thin stems. What we need from the research is a continuous supply of new, improved varieties that are resistant to plant diseases and high yielding in order to sustain our production.*

In both districts, newly established CSBs have re-introduced preferred local varieties from genebank collections and well-established CSB in similar agroecological areas. Although the FGD participants in Heexosa felt that traditional varieties did not perform well for high yield, some farmers expressed interest in gaining access to durum wheat varieties with important cultural values and appreciated the reintroduction of lost durum varieties in the face of high genetic erosion (75–100% loss) in the central highlands [103–105].

### 5.2. Seed Availability

Seed availability is adequate when farmers can source enough seed at the right time to meet their needs from available sources [19]. In post-disaster contexts, seed security studies typically find that even when farmers' own seed saving is reduced, seed continues to be available from other sources, especially local markets [52,106,107]. Exceptions to this are often linked to disease outbreaks, especially for vegetative crops, or disruptions in the functioning of social networks, markets/road networks, or the formal seed system (for certified seeds) [30,108,109]. Understanding seed availability thus starts with gauging the relative importance of different seed sources. Our survey shows that farmers in both districts overwhelmingly rely on farm-saved seeds, both for their major crop and secondary crop (Figure 3). Social networks are the second largest source for the dominant crop in both districts, but in Heexosa, where farmers rely more strongly on improved varieties, the public seed sector is almost on par with social networks.

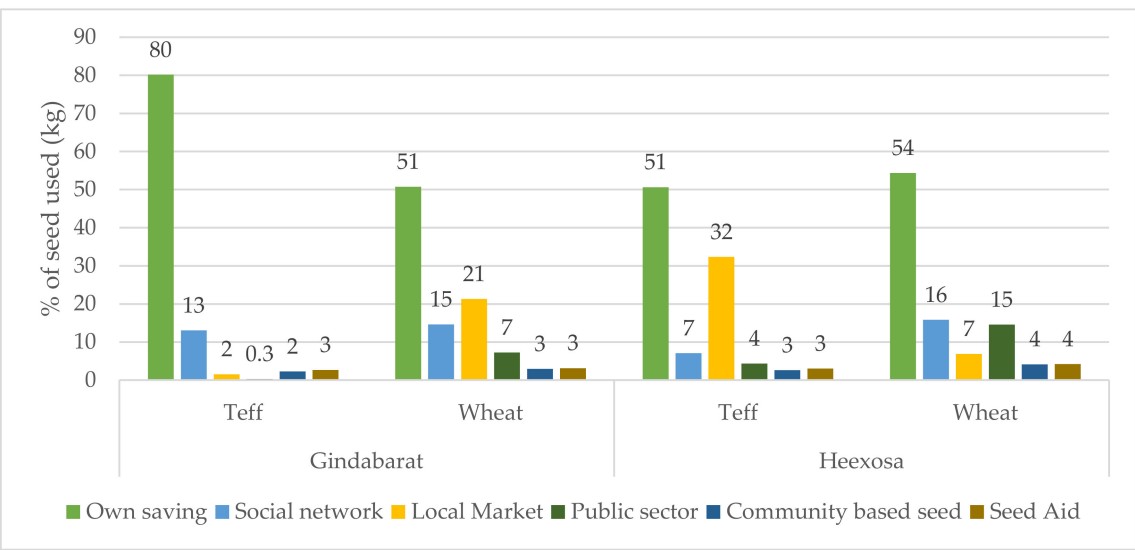

**Figure 3.** All seed sources as percentage of all seed used for teff and wheat in Gindabarat and Heexosa districts in 2017/2018 planting season, respondent households (*n* = 223 teff growers and *n* = 28 wheat growers in Gindabarat, and *n* = 209 wheat growers and *n* = 60 teff growers in Heexosa). The seeds sources included (1) own savings; (2) social network (exchange with relatives, neighbors and/or friends); (3) local markets; (4) public sector (parastatal seed enterprises, associated PMCs, agricultural research centers and district agriculture extension bureaus; (5) community-based seed groups (SPCs and CSBs); and (6) Seed Aid (emergency seed relief programmes in Heexosa and Afoosha self-help group in Gindabarat). Direct Seed Marketing (DSM) represented a negligible volume of seed in Heexosa, and was excluded from the figure.

The high reliance on own-saved seed is in line with other studies of cereal seed systems in normal conditions [32,110–112]. FGD participants in both districts indicated that they consider their own saved seed to be the most reliable seed source. Even for improved varieties of wheat in Heexosa and the one commercial variety of teff in Gindabarat (Quncho), farmers primarily use own-saved seed, relying on social networks and the public sector mainly for seed renewal purposes. This is consistent with a study of major wheat growing areas in Ethiopia, which showed that about 84% of the farmers depend on recycled seeds while only 14% used new seeds [113].

Some seed security studies show that local markets play a major role for many crops in post-disaster areas [71,114]. In our study, this is only true for the secondary crops in each district, for which local markets were the second largest source (approx. 20–30% of seed). The secondary crops are grown by a minority of households, and not necessarily every year; FGD participants explained that farmers often invest less effort in seed saving for these crops, relying instead on the local market.

Community-based and seed aid contributed less than 5% of the quantity of seeds in both districts (Figure 3). In Gindabarat, there were no SPCs, while in Heexosa recently established cooperatives produced non-certified seed, which they sold locally. This included 15% of certified seed they produced for the public sector, which they can lawfully retain, as well as "QDS seed", though in practice this was not quality controlled. There was also a CSB in one ganda of each district that produced seeds of traditional varieties that were not common in the district. In Heexosa, this focused on traditional wheat varieties (i.e., durum wheat) that were almost entirely lost due to displacement by improved bread wheat varieties over the past five decades [104].

In Gindabarat, seed supply by agrodealers or seed agents is absent. In Heexosa, we encountered a few direct seed marketing (DSM) agents supplying wheat seeds from the public seed enterprises, but only a few farmers (*n* = 5) in our survey bought seed from these agents. DSM was introduced in Ethiopia in 2011 to enable public and private seed producers to directly assess seed demand and supply adequate quantities of seed in convenient locations using either their own sales staff or hiring private agents [115,116].

According to FGD participants, the main seed security challenges relating to availability in the two districts was the lack of adequate and/or timely supply of certified seeds. In Heexosa, certified seeds produced by the public sector were insufficient or distributed late, whereas, in Gindabarat, certified seed use was limited to small quantities of Quncho (0.3% of seed) and wheat varieties (7% of seed) that PMCs receive from the Ambo Farmers' union and sell to farmers. The almost negligible contribution of the public sector to teff seed supply in Gindabarat is much lower than the national average of 10% [110], and FGD participants emphasized that the demand among farmers for certified seeds like Quncho is much higher than the supply.

Although the overall frequency of calamities is perceived to be low in both districts, FGD participants pointed out that drought, flood, and plant diseases (e.g., Ug99) have been increasing in recent years. As a result, FGD participants in Gindabarat expressed their desire for external support to establish a local grain/seed reserve suitable for long-term seed storage to ensure local seed availability during disasters. They discussed this in connection with Afoosha self-help groups that provide donations of grain/seed to poor households affected by socioeconomic and environmental disasters, as well as CSBs that provide low-interest seed loans. They explained that Afoosha is based on an indigenous long-term seed storage practice called *dilbii* (grain/seed reserve) in which rich farmers *Abba dilbii* ("owners of grain/seed reserve") who saved teff seeds/grain up to seven years in well maintained *gotooraa* gave seed/grain to poorer households for free or as credit. *Gotooraa* is the name in the Oromo language for medium and large sized cylindrical or rectangular granaries made from bamboo or sticks and built on a bed having four forked support poles. They are plastered with mud and dung and dried before use for grain/seed storage. *Dilbii* has disappeared due to successive land redistribution programs and increased poverty but has been reinvigorated by Afoosha in Gindabarat, where the practice is widespread. In Heexosa, FGD participants felt that increasing the SPCs' annual seed production and supply at the community level would be more appropriate than establishing a seed reserve due to the shorter storability of wheat seeds.

Finally, despite the efforts of the CSB in Heexosa, the availability of adaptable durum wheat varieties is still very limited in the district. This is also the case of improved durum wheat varieties that have been developed by the public sector [117] but are not multiplied or made available to farmers [113].

### 5.3. Seed Access

Seed access refers to farmers' ability to acquire seed, whether it be with cash or through exchange, loan, or social networks [19]. In addition, CGIAR [26] identifies seed access as depending on extension and seed dissemination/delivery channels (e.g., transportation and distance) and sufficient information/awareness about how and where to get quality seed, as well as information on prices. Problems with seed access tend to be among the most common challenges facing farmers in emergency contexts, due to acute problems such as loss of financial resources or assets and insecurity/inability to travel to markets, while also exacerbating chronic vulnerabilities experienced by specific socio-economic groups [30,102,118]. Insights from the adoption literature on factors associated with the use of improved varieties also provide a useful backdrop for assessing the access dimension of seed security.

Here we focus on access to seeds that were considered expensive by farmers: recycled Quncho seed that is obtained through exchange or purchase via social networks in Gindabarat and certified wheat seeds sourced from the public sector in Heexosa. Our survey results show that in Gindabarat, Quncho represented a higher share of total teff seed volume for male heads of household (MHH) compared to female, self-reported wealthier farmers compared to medium and poor, and for younger farmers compared to older farmers (Table 4). In Heexosa, there was relatively little difference between gender, wealth, or age groups in terms of the share of wheat seed volume represented by improved varieties (Table 4). However, more substantial differences among groups were observed in farmers'

use of certified vs. recycled seed for improved varieties (Table 5). Compared to wealthier farmers, poor farmers used less certified seed and recycled seed for longer, with 13% doing so beyond the maximum of five years recommended by research [119]. Interestingly, relatively *more women* (FHH) used certified seeds than men (42% vs. 27% of respondents), and no FHH recycled seeds for more than five years. As described below, there are several factors that explain these trends: purchasing power, access to information, and privileged positions within government rural development programs, and how they are differentiated according to gender, wealth status, and age.

**Table 4.** Percent seed volume represented by improved varieties for the major crop in Gindabarat (Quncho) and Heexosa (Wheat).

| | Gindabarat | *n* | Heexosa | *n* |
|---|---|---|---|---|
| **Gender** | | | | |
| MHH | 33% | 190 | 90% | 172 |
| FHH | 25% | 33 | 83% | 37 |
| **Wealth status** | | | | |
| Poor | 25% | 30 | 83% | 28 |
| Medium | 32% | 182 | 92% | 166 |
| Rich | 36% | 11 | 71% | 15 |
| **Age** | | | | |
| Young <45 years | 40% | 118 | 86% | 117 |
| Old ≥45 years | 23% | 105 | 93% | 92 |

**Table 5.** Percent of farmers using certified and recycled wheat seed in Heexosa.

| | Certified Seed (Changed Annually) | Recycled 2–5 yrs | Recycled >5 yrs | *n* |
|---|---|---|---|---|
| **Gender** | | | | |
| MHH | 27% | 65% | 8% | 172 |
| FHH | 42% | 58% | 0% | 37 |
| **Wealth status** | | | | |
| Poor | 22% | 65% | 13% | 28 |
| Medium | 30% | 64% | 6% | 166 |
| Rich | 50% | 50% | 0% | 15 |
| **Age** | | | | |
| Young <45 years | 33% | 59% | 8% | 117 |
| Old ≥45 years | 26% | 69% | 5% | 92 |

High seed/grain price for both Quncho in Gindabarat and certified seed in Heexosa was identified by FGD participants as limiting factors for poor households. In Heexosa, this is one of the main reasons that poorer households recycle improved seed for longer. As described by the FGD participants, poor farmers do not have access to newly released wheat varieties for the first couple of years until enough recycled seed of the new variety is available in their communities through SPCs and social networks at more affordable prices:

> *In our districts, all gandas have at least one PMC. All of us are supposed to buy certified seeds and other inputs from the government enterprises at the PMC shops. The price of 100 kg of certified wheat seeds from the PMCs [1350 ETB] is almost twice the price of our grain produce [800 ETB]. [ETB: Ethiopian birr; 1USD = 40ETB]. Not all of*

*us access because of the high price. ( . . . ) Renewal of seed or getting seeds of a new, improved variety is extremely challenging because there are not enough seeds. During the first few years, only model farmers and out-growers get the seeds of the newly arrived variety. These seeds are much more expensive than seeds of older varieties. ( . . . ) Timely access to seeds of a new variety is not possible. The positive thing, though, is that out-growers/model farmers sell at a lower price [1200 ETB] than the PMCs. For them, it is still profitable compared to the grain price.*

While high price is one limitation, this explanation also reveals that model farmers have preferential access to certified seed compared to other farmers. The FGD participants also underlined that poor farmers are rarely recruited as model farmers. Key informants explained that this is due to poor farmers' small landholdings and assets, which limit their ability to participate in seed multiplication and dissemination. They also pointed out that political allegiance was used by the district to select model farmers. This is consistent with Hailemichael and Haug [120]'s study of the extension system and advisory services in eight districts of Ethiopia, which argues that political allegiance is a major factor influencing the selection of model farmers, favoring wealthier farmers aligned with the government, and giving them privileged access to information, technology, and new skills, to the exclusion of other farmers. It is also consistent with political extension studies that view the model farmer approach to agricultural extension in Ethiopia as a historical continuation of the exploitative power relations between farmers and the regime [121–123].

The case of Quncho in Gindabarat shows that high grain/seed price is relevant not only for seed accessed through the formal seed system but can also play a role even for seed accessed through social networks. Due to Quncho's high market value, FGD participants explained that it is expensive to obtain Quncho seeds/grain using cash or in exchange against other crops/varieties of equivalent value. They described that this limits access for poor farmers, who have large families compared to their landholdings, and prefer to produce another teff variety, Daaboo, for household consumption. Thus, access and use of Quncho was more common among wealthier farmers.

Our finding that younger farmers also use more Quncho than older farmers was unexpected because farmers under the age of 45 years tend to lack adequate farmland and/or be considered poor because few of them participated in the last land redistribution in Ethiopia following the fall of the military government in 1991. It is therefore surprising that they are willing to pay the price for pure Quncho seed. According to FGD participants, this was because the younger farmers with limited landholdings preferred to grow Quncho for its market price and purchase cheaper grains such as maize and sorghum for home consumption. This strategy allows them to secure more food grain than growing Daaboo, but the *buddeena* made from these crops is considered inferior to that made from teff, and its consumption is considered a sign of poverty. A similar strategy of selling high-value improved wheat to purchase maize and sorghum was also described by younger participants of the FGDs in Heexosa. Thus, younger farmers were willing to sacrifice food quality for economic gain.

While limited purchasing power is a constraint to seed access for all resource-poor groups, our study reveals dynamics related to access to information that are specific to gender, with surprising contrasts between the two districts. In Gindabarat, FHHs self-reported more frequently as poor than their male counterparts (27% of FHHs vs. 11% MHHs), and this may be one reason that they also use less Quncho seed than MHHs. However, FGD participants also agreed that men were better represented in agriculture and rural development related meetings and trainings provided by local extension, which enabled them to get more knowledge about improved seeds than women. Timely and reliable information about farmers who have good quality surplus Quncho seed was also exchanged at these gatherings, giving men an advantage in sourcing Quncho seed. Women FGD participants mentioned that most of them were not members of PMCs and were not recruited as model farmers. They also spoke about a lack of time to attend agricultural extension meetings and trainings when they were invited. This result is

consistent with other studies that have found that 'non-model' and/or women's limited access to agricultural awareness creation platforms influences their access and use of agricultural technologies [123,124].

The situation for FHHs was markedly different in Heexosa, where women used more certified seed than men and recycled it within the recommended time frame (Table 5). Adoption studies show that the decline in wheat productivity can be improved by using new certified seeds compared to older recycled seeds [125,126] and that frequent seed renewal by smallholder commercial farmers shows their productive behavior [112]. Knowledge on the use of agricultural technology is created mainly through access to information [127], and this is a strong indicator of women's empowerment [128]. Interestingly, female FGD participants explained that compared to men, FHHs in Heexosa had equal access to information and agricultural inputs, including certified seeds. FHHs also had similar wealth status (16% FHH vs. 13% MHH self-reporting as poor) and were well represented as model farmers. The women FGD participants highlighted unexpectedly positive empowerment of FHH, and their related access to improved agricultural technologies:

> *Unfortunately, all of us are on our own i.e., we are widows and divorcees. ( . . . ) We do everything that most men do in farming. In the past, women, including widows and divorcees, were not considered equal to men. Now, we have more freedom and voice. We equally participate in meetings, trainings, and access inputs as men. We express our ideas in public gatherings. In recent years, we are also privileged to sometimes get priority over men for inputs and trainings due to our active engagement, which authorities appreciate. We learnt new techniques and gained skills in agriculture. We have better savings; some of us have saved between 70,000 to 100,000 ETB. We have full control over our incomes and resources. We hire labor and rent land to expand our production. In fact, some of us are better than many male farmers.*

This is a striking account considering the patriarchal culture in Ethiopia as well as socioeconomic and political marginalization of women in all sectors, including agriculture [129,130]. Indeed, it seems to reflect an important change in agricultural technology use over time, as Tiruneh et al. [124] found that 20 years ago, FHHs in central Ethiopia used improved seed 50% less than their male counterparts. Although it requires further investigation, FHHs' high empowerment and access to agricultural inputs and positions was explained by key informants in terms of "effective" implementation of the government's decentralized extension program, citing among other things that the posting of female development agents in every ganda has been very useful for agricultural technology dissemination. In addition, they explained that the strong presence of externally supported development projects in the district has led to a significant push for a gender-sensitive approach to agricultural development. The long history of agricultural development interventions, combined with donor requirements for gender mainstreaming, therefore seems to have created opportunities at least for FHHs in Heexosa, in contrast to the situation in Gindabarat, where external agricultural development actors are largely absent.

It is important to note, however, that the FGD participants emphasized that married women did not benefit from the same kind of empowerment as FHH did. As they explained: "They are still under the control of their husbands. They do not go out and participate in meetings and trainings. They are powerless. There is a big difference between married women and us".

### 5.4. Seed Quality

Seed quality refers to the physical, genetic and physiological properties of the seed, including germination, vigor, varietal purity, and freedom from disease and impurities, and is crucial for farmers to establish robust plant stands and harvest higher yield [19,131,132]. Problems with seed quality are among the major challenges facing farmers in emergency contexts. This is mainly due to poor seed quality management practices among seed traders, NGOs, and other actors involved in seed relief [114,133], as well as chronic is-

sues that smallholder farmers experience with pests and diseases, seed handling and storage [134–136].

Here, we examine farmers' perception of the quality of seeds they obtain from different sources and the storage facilities they use. Focusing on seeds obtained during the 2017/2018 growing season, we asked respondents to rank the quality of the seeds used for each variety of their major crop (i.e., seed lot) as "good" and "not good" and triangulated this with qualitative assessment by farmers in FGDs. Farmers rated seed as "good" if they had no weeds, debris, varietal mixture, had good germination and were free of insect damage; and "not good" if most of these seed quality features were lacking. Our survey shows some marked variations in seed quality between seed sources and districts (Figure 4). In both districts, community-based seed (from SPCs and CSBs) was rated by farmers as the highest quality, with less than 10% of seed lots considered "not good". FGD participants in Heexosa explained that the SPC members pooled together knowledge and experience, and the trainings they received from experts from the public seed enterprises, research, and bureau of agriculture helped them to maintain good quality standards for the seeds they produced. In addition, CSB members spoke in the FGDs about how the CSB technical committee assessed quality based on information they gathered on preharvest handling and through visual inspection when members paid back their seed loans. We also observed a good storage facility that one SPC had built with external support. A study by Sisay et al. [77] supports farmers' assertions that organized seed producer farmers maintain higher quality standards than individual households. On average, 94% of all cereal seeds produced by farmers' groups for the Ethiopian Seed Enterprise in the 2009/2010 season were approved as certified seeds [137]. While these findings show that community-based seed has good quality, there are recent studies that report infrastructure challenges and poor seed handling practices among SPCs [138,139]. Moreover, these seed sources are marginal in terms of seed volume in our study areas (Figure 3).

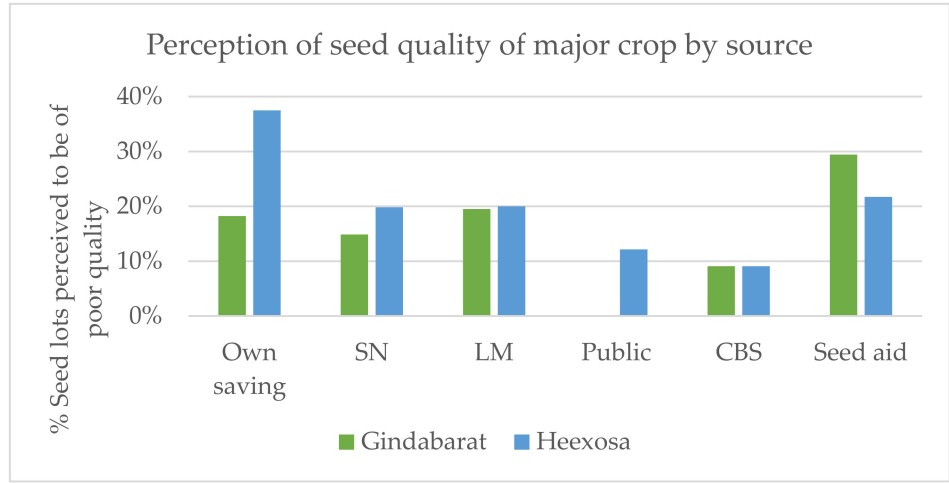

**Figure 4.** Percentage of respondents' rating of seed quality (germination/vigor, physical purity, varietal purity and seed transmitted diseases or sanitary conditions) for seeds they accessed from different sources (Own saving, SN = Social seed network, LM = Local market, CBS = Community-based seed producers, Seed Aid = emergency seed relief programmes in Heexosa and Afoosha self-help group in Gindabarat) during the 2017/2018 planting season (*n* = 605 teff seed lots in Gindabarat and *n* = 758 wheat seed lots in Heexosa). "Seed lot" is defined in this paper as the seed from a specific teff or wheat variety that was planted by a household in the 2017/2018 season. Only one farmer in Gindabarat rated the quality of seed from the public sector, so this was excluded.

In line with other studies [30], farmers reported quality problems with seed aid, particularly in Gindabarat (27% of seed lots), where the traditional Afoosha self-help system was the main source. The Afoosha grain reserve has relatively poor quality because it is established through contributions of food grain (rather than seed) from different



families, though many aid recipients use it for planting. In addition, the Afoosha do not typically have good seed storage facilities. In Heexosa, key informants explained that seed aid is provided by NGOs and international research institutions (i.e., CIMMYT), who source the seed primarily from SPCs and public enterprises, being one of the main customers of for these groups [138]. Afoosha are present in Heexosa but are not involved in seed/grain aid.

Own-saved seed is the major seed source in both districts, and therefore seed quality problems for this source are of particular importance. There were seed quality problems for own saved seed in both districts, but this was particularly high in Heexosa (37% of seed lots). FGD participants in both districts related seed quality problems mainly to the mixing of varieties between harvest and processing stage, as well as problems in seed storage. In Heexosa, FGDs explained that varietal impurities were caused by the use of communal combines, which were used to harvest plots of neighboring farmers with insufficient cleaning in between. Storage problems for wheat were chiefly caused by seed-borne fungi and granary weevils. In both districts, mixing of white and brown-seeded teff varieties was the major problem with varietal purity due to the commercial value associated with the seed color. Otherwise, mixtures between local varieties were not considered problematic. The main causes mentioned were heavy rains and run-off after planting that transports seeds and seedlings from one field to another, as well as poorly cleaned winnowing fields and seed storage. The main storage problem was high moisture levels in the seed storage caused by insufficient drying of seed after untimely rain during harvest and winnowing.

One reason that the quality of own-saved seed in Heexosa was perceived to have more problems than in Gindabarat could be differences in the inherent storability of the crops. Due to its small seed size and resistance to insect pests, teff has good viability for up to five years if stored following a proper drying [140,141], while wheat can typically not be stored for more than two seasons due to infestation by granary weevils and/or fungal diseases [32]. In addition, we found significant differences in seed storage practices (Table 6). In both districts, survey respondents stored seeds inside their homes, but in Heexosa, woven polypropylene bags were used to store 90% of seed lots, with chemical insecticide applied to increase storability. In contrast, in Gindabarat *gotooraa* played a much more important role than in Heexosa. Gotooraa is perceived to have a better aeration, and only 16% of teff seed lots stored in *gotooraa* were reported as "not good" in Gindabarat, compared to more than twice as many (37%) wheat seed lots stored in polypropylene bags in Heexosa. In Heexosa, FGD participants explained that *gotooraa* has been abandoned as households have increasingly adopted an urbanized way of life and thus do not have enough space inside their homes to build a bulky *gotooraa*. Furthermore, increased grain theft discouraged farmers from building *gotooraa* outside their homes, except in predominantly Muslim gandas where theft is uncommon. A decade ago, 66% of farmers were using the facility in northwest and central Ethiopia [32]. At the same time, farmers have not yet adopted hermetic bags that are effective for seed storage [142] due to lack of awareness and high price [143,144].

**Table 6.** Table Storage facilities most used per district. Data is presented as the % of seed lots produced from own-saved seed. "Seed lot" is defined in this paper as the seed from a specific teff or wheat variety that was planted by a household in the 2017/2018 season.

| Storage Facility | Heexosa (Wheat) | Gindabarat (Teff) | Total |
|---|---|---|---|
| Woven polypropylene bags | 89.5% | 41.5% | 64.2% |
| Gotooraa | 0.5% | 33.0% | 17.7% |
| Woven polypropylene bag with inner liner | 3.3% | 13.4% | 8.6% |
| Jute | 0.5% | 6.5% | 3.7% |
| Other ‡ | 6.3% | 5.6% | 5.9% |

‡ Plastic bag, metal/plastic drum, earthen/clay pot, gourds, loose in a room and community storage facility such as CSBs.

In Heexosa, the quality of seed lots sourced from the public sector seeds was high, with only 12% of lots reported as "not good", nearly on par with the community-based seed (Figure 4). In Gindabarat, only one farmer rated the quality of seed from the public sector, so this was excluded from the analysis. Nonetheless, our qualitative information gives a different picture. FGD participants in both districts asserted that the quality of their own-saved seed was equal to or sometimes even better than certified seed. This is consistent with studies in Syria and Ethiopia, which found that about 90% of farmers are satisfied with the quality of own saved seeds for cereal crops [32,98,145]. Moreover, male FGD participants in Heexosa spoke with utter disappointment about the certified seeds they accessed:

> *We want to tell you that the seeds we buy from PMCs have no good quality. They mix seeds from the present season with unsold seeds carried over from the previous season, seeds produced in different agroecologies, as well as seeds of different crop species/varieties, and sell to us. Sometimes, we found barley in a package of wheat seeds that we bought. The wheat seeds we purchased from them did not perform uniformly when we sowed in the field. They were like our fingers [a farmer shows different length of his fingers]. They did not have equal height, awn types, and panicle size.*

## 6. The Potential of the Ethiopian Seed System Development Strategy to Meet Demands

The results and analyses presented above testify to the widespread seed insecurity in both the commercially oriented wheat-centered seed system in Heexosa and in the subsistence-oriented teff-centered seed system in Gindabarat. In this section, we ask to what extent does Ethiopia's shift from a linear model of seed sector development to a pluralistic approach holds potential to improve farmers' seed security? We address this question by analyzing the relevance of the PSSDS' priority interventions (for each seed system and their cross-linkages) in relation to our empirical findings on seed security (above), and examine how the underlying functioning of the seed systems, as revealed by our analysis of seed sector actors' roles and performances (Table A1), pose constraints and opportunities for the PSSDS' implementation.

### 6.1. Informal Seed System

The informal seed system provides most of the seed volume for the major crop in each district, mainly from own-saved seeds and social networks, with the local market being more important for the secondary crops. The PSSDS includes several priority interventions to strengthen the informal system (Table A2), including:

- Improving access to locally adapted varieties by strengthening coordination between farmers, research centers, and genebanks for re-introduction of lost varieties, selection of locally adapted varieties, and by improving access to germplasm for participatory varietal selection and breeding;
- Increasing the diffusion of local varieties through innovative marketing networks (seed fairs, field days, open markets) and through investment in CSBs, including allocating gene funds from access and benefit–sharing agreements;
- Setting up a national system for seed provision during emergencies to improve emergency response, including the establishment of a national seed reserve, creating an independent institution to lead seed security assessments and interventions, and strengthening quality control measures for emergency seed;
- Improving awareness, skill, and infrastructure to improve farmers' production and management of good quality seed.
- *Cross-linkages—informal and formal:* engaging farmers, agricultural research, and regulatory authorities in participatory varietal development and release to ensure varietal suitability for farmers; supporting farmer-genebank linkages using the gene fund to compensate farmers' management of local genetic diversity.

These are all relevant to addressing key seed security issues identified in this study, such as the loss of traditional durum wheat varieties, interest in strengthening teff seed re-

serves (Gindabarat), and challenges in the perceived quality of some informal seed sources, such as Afoosha seed aid (Gindabarat) and own-saved seed (especially Heexosa). However, our study shows that interventions to improve the informal seed system have largely been left out from current government-seed related programs, with the only supports being made through NGO interventions backed by international donors, e.g., CSB projects financed by western NGOs and "crowdsourcing" to engage farmers in participatory testing and dissemination of open-pollinated local and improved varieties through the ISSD program (Table A1; [138]).

Our key informant interviews show that government- and NGO-led interventions in both districts have been riven with conflicts (Table A1). On the one hand, district agriculture bureaus promote the use of 'the full package' (improved varieties, chemical fertilizer, pesticides, and improved agronomic practices) as a means to achieve higher yield. We found that the district development agents were doing everything possible to convince farmers to adopt the package, as their salary and benefits are related to fulfilling the district's targets for adoption. This finding is consistent with studies of the Ethiopian agricultural extension system during the Sasakawa Global 2000 program in the 1990s [146,147]. On the other hand, NGOs focus on community-based seed production, emphasizing the superiority of traditional varieties for higher yield stability, low-cost input, better nutrition and adaptation, and encouraging farmers to diversify crop production. This has created confusion among farmers in areas where NGO projects were implemented. Our interviews with key informants suggest that the lack of coordination is partly due to a lack of awareness about the PSSDS and the mandates it prescribes among stakeholders at local level, which calls for attention by all actors.

This disjuncture at the local level mirrors a conflict at the national level that has existed for decades between the Ethiopian Institute for Agricultural Research and Ethiopian Biodiversity Institute concerning their respective mandates [63], and ideological differences regarding the use of Green Revolution technologies versus local varieties adapted to low input agriculture [148]. Since the 1990s, the Institute for Agricultural Research has asserted that the main task of the Biodiversity Institute should be limited to ex situ conservation and making germplasm available for formal breeding, arguing that development of high yielding varieties is critical for food security, whereas the Biodiversity Institute, who has supported community-based seed management initiatives together with allied civil society organizations, has insisted that promotion of diverse varieties, and specifically locally adapted landraces, is critical to strengthen farmers' seed systems in face of recurrent drought and genetic erosion. Both institutes participated in the development of PSSDS and advocated for their respective approaches, but thereafter have continued to implement their programs as before, without making adjustments for complementarity between value-chain components of each seed sector and integrating their activities, as set out in the PSSDS. The one exception is the ISSD program, in which diverse stakeholders have been involved in efforts to properly implement the cross-linkages identified in the PSSDS, e.g., linking farmers with agricultural research for crowdsourcing and participatory variety selection [149]. More generally, this lack of implementation of the PSSDS is due to the authoritarian nature of the Ethiopian state itself. In his study of seed policy in Ethiopia, Beko [150] finds that although the government often seeks stakeholders' input for policy making, this is mostly done to meet official procedure or as a formality. In practice, only policy provisions that are in line with the government's political objectives are implemented, with the aim to maintain control over farmers and secure political allegiance [74,151,152].

This lack of attention to the informal seed system in Ethiopia's agricultural development programs has major implications for seed security. For example, the finding that over a third of own-saved wheat seed lots used by farmers in Heexosa were rated as poor quality is concerning. Aside from some technical trainings and awareness creation provided to a limited number of farmers who participated in out-grower schemes, SPCs, and the CSB, there are nearly no interventions to help strengthen household-level seed handling and storage (Table A1). The combination of deteriorating traditional seed storage practices

and incomplete modernization—affecting the quality of the most important seed source used by farmers in Heexosa—deserves concerted investment at scale by government and NGOs alike.

The PSSDS provision to develop a national seed emergency system has yet to be implemented but responds to the desire for a reliable local teff seed reserve, expressed by the FGDs in Gindabarat. The seed security literature identifies direct seed distribution and market-based approaches such as vouchers and seed fairs as typical emergency seed aid interventions [30]. Our findings suggest that there is an opportunity for the development of a seed emergency system building on existing community institutions such as Afoosha and CSBs, as proposed by the FGDs. Given the working principles such as trustworthiness and altruism [153], Afoosha is a strong, cohesive force at the community level and has effectively reinvigorated the practice of *Abba dilbiis*. However, unlike *Abba dilbiis*, our findings indicate that seeds/grain from Afoosha are of low quality as the seeds are from grain reserve. This, therefore, requires technical and management solutions such as separating varieties, proper drying of seeds, having a proper warehouse and maintaining seed stores, recommended also for other seed aid actors [114,154]. Although Afoosha seed aid is widespread in Gindabarat, its scope at national level is not clear, which calls for further study. Seed reserves would be less appropriate for crops with short storability like wheat.

To finance efforts to strengthen the informal seed system, the PSSDS proposes to establish a fund derived from monetary benefits that the federal government expects to gain from international Access and Benefit-Sharing agreements under the International Treaty on Plant Genetic Resources for Food and Agriculture (ITPGRFA) and the Nagoya Protocol of the Convention on Biological Diversity (CBD). Thus far, some projects for on-farm management of genetic resources have been funded by the Benefit Sharing Fund of the ITPGRFA [155,156] and the CBD have supported development of national policy and legal frameworks for the implementation of Nagoya Protocol [157]. However, monetary benefits have not been generated from the small number of bilateral access and benefit-sharing agreements established under Nagoya Protocol to date and such funds are likely to remain limited [63]. Investment to improve the informal sector should thus be based on other more reliable funding sources (e.g., over the regular agriculture budget).

*6.2. Formal Seed System*

Farmers in both study districts show a clear interest in improved varieties but rely mainly on informal channels to source seed. In Gindabarat, this is due to the near total absence of formal seed system actors in the district, whereas in Heexosa the system is well established but suffers from ineffective performance.

The PSSDS recognizes several bottlenecks in the formal seed system and proposes a comprehensive approach to "bring about a holistic transformation" of the system (Table A2). Some of the main interventions proposed are:

- Improving the development of adapted crop varieties by strengthening the coordination of federal and regional research centers, promoting participatory plant breeding, and establishing a body independent of research institutions to oversee variety registration, release, and protection;
- Increasing the volume of certified seed by addressing inefficiencies in the value chain (including improving the accuracy of seed demand estimation and delineating responsibilities for the production of each seed class) as well as by increasing the capacity and number of out-growers;
- Improving the timeliness of certified seed supply through DSM and by replacing government price setting with open pricing to reduce delays due to excessive bureaucracy;
- Strengthening access of resource-poor farmers, especially women, to certified seed through credit and savings schemes;
- Improving seed laboratories' capacity for seed quality inspection and testing by building their technical capacity, infrastructure, and equipment, as well as increasing the number and remuneration of technical staff.

- *Cross-linkages—formal and intermediate*: institutionalizing out-grower and agrodealer schemes by establishing contractual agreements between public seed enterprises and community-based seed producers (e.g., SPCs and PMCs) for certified seed production; improving community-based seed producers' access to basic and first generation certified seeds for production of QDS.

Our findings from Heexosa indicate that several investments have been made in the last decade that align with the strategy. This includes increases in the production of certified seed by strengthening the Oromia Seed Enterprise and expanding the number of out-growers organized in cooperatives and commercial cluster groups [138,139]. Contractual agreements have also been established between seed producer cooperatives and public seed enterprises for seed multiplication, although many breaches in these contracts have been reported, linked to price setting and capacity [158]. Since 2011, ISSD, together with Ministry of Agriculture and the Agricultural Transformation Agency, has also piloted DSM, which has now been scaled out to 313 districts nationally [149,159]. Key informants in Heexosa indicated that this has reduced delays in seed supply to farmers, and according to Alemu et al. [149], efficiency has been increased especially by reducing costly rates of seed carryover in store by as much as 85%. From the perspectives of development actors, the expansion of DSM is seen as partial liberalization of the seed sector, a step in the right direction to transform the formal seed system [116,159,160]. A regional seed laboratory has been established in Asella, taking over most of the responsibility for field inspections and certification from the national laboratory. The regional and federal agricultural research centers have been assigned specific responsibilities for variety development and adaptation research (Sinana on durum wheat and Kulumsa on bread wheat), albeit with some overlaps.

Despite these efforts, important barriers still exist that underly several of the seed security problems identified in our study. One key issue is the need for better-adapted improved varieties, reflected in the general lack of improved varieties of teff, as well as the need for better disease resistance in wheat. Although teff is the most frequently grown staple food and biggest cash crop in Ethiopia [161], it is generally considered an "orphan crop" receiving little attention from international research and the donor community [162,163]. At the time of our field research, there were 37 improved teff varieties listed on the national crop variety register [117] and one newly released brown seeded variety [164], but only five of them were adopted nationally for extensive cultivation, with two of these (Quncho and Tseday) accounting for 90% of all teff seed production [110,165]. The main reason that the other varieties are not distributed is the lack of farmers' preferred traits such as lodging and biotic/abiotic stress tolerance, non-shattering, and higher yield [162]. In Gindabarat, this challenge at the national level is compounded by the lack of local research on the adaptation of released varieties. The district agriculture bureau's interventions are limited to sporadic theoretical trainings for model farmers on the importance of agricultural technology packages, without corresponding investment in seed production and supply of improved varieties. By and large, most FGD participants viewed certified seed from the formal system as risky and government seed supply as something they cannot rely on. In effect, Gindabarat can be considered an "orphan district" in terms of the formal seed system.

There has been more investment in wheat and high potential districts such as Heexosa. Kulumsa Agricultural Research Center has released approximately 70 improved bread wheat and durum wheat varieties [166], but disease resistance—especially to Ug99—is a global challenge [99,167]. It is therefore not surprising that farmers in both districts expressed challenges with this. Nonetheless, as with teff, it is also the case that many released wheat varieties are sitting on the shelf. According to key informants and recent project reports [168], some of the new varieties that have very farmer preferred traits, e.g., the variety Kingbird with superior disease resistance [169], have been quickly adopted and spread through informal channels by model farmers who get the seed in adaptation trials. Yet, distribution through formal channels has lagged for many varieties due to bottlenecks in the regulatory system. As with teff, some key informants indicated that there

are improved wheat varieties that are not distributed due to the lack of farmer-preferred traits, pointing to the poor involvement of farmers in the breeding process. The lack of better-adapted varieties is therefore compounded by constraints in variety deployment (including variety promotion, seed production and dissemination), which is a major issue for many crops in most developing countries [170,171].

The PSSDS' plans to promote participatory plant breeding holds potential to improve the suitability of released varieties. Indeed, Quncho was developed through participatory plant breeding and multi-station variety selection on black soils, which helped to successfully incorporate farmers' criteria [172,173]. Furthermore, the provision to establish an independent body for registration and release and to increase the efficiency of the registration and release process. However, it is less clear to what extent these and other investments will be made in "orphan districts" like Gindabarat.

Another key problem is the shortage of certified seeds and delays in supply. Key informants from Kulumsa Agricultural Research Center (KARC) explained that this is partly due to lack of resources (funding, land, infrastructure, and technical capacity), which constrains the production and distribution of early generation seeds (EGS), and the low number and capacity of the federal and regional seed enterprises. In addition, the seed recovery rate from out-growers is generally low, as out-growers often retain more than the 15% share they are entitled to [111,112]. As a result, only limited quantities of popular varieties are produced. Perhaps even more critical are problems with seed demand estimations and quota allocations that are carried out under the oversight of the Ministry of Agriculture. In Heexosa, none of the district agricultural development agents we interviewed had confidence in the seed demand information they collected, citing mistrust in the information provided by farmers, the failure to collect current data due to the lack of transport, and high demands of other tasks, among others. This is consistent with Hailu et al. [174] who found that the poor performance of Ethiopia's agricultural extension system was explained by limited synergy and partnership among actors, poor motivation and competence among development agents, and insufficient resources for their mobility. In Gindabarat, key informants explained that the very limited amounts of certified Quncho seed that reach PMCs often do not correspond to seed demand estimates that the district development agents submitted. They blamed this on the top-down manner in which the formal seed supply system is governed, with minimal participation of local government.

To be more effective, seed demand estimation should be made directly by public seed enterprises and other private seed producers rather than by the Ministry of Agriculture. In this sense, DSM seems to be a viable alternative to improve the performance of the formal seed supply for self-pollinating cereal crops in Ethiopia. However, to do so will require the Ministry of Agriculture to relinquish direct control over parts of the seed supply system, focusing instead on coordination and regulation. Whether there is political will to do so is an open question, considering that seed supply in Ethiopia has to date been politically driven. For example, the government has used input provision as a way to control farmers and secure their political support [120,175,176]. To maintain their dominance of the seed sector, the government has also curtailed the role of private sector actors by using market disincentives such as price setting or limiting areas of operations for seed marketing [74,177]. In practice, the government has been skeptical towards the private sector, despite the many policy documents promising to strengthen their involvement [74].

One of the main goals of any formal seed system is to provide seeds of verified quality, yet FGDs in both districts pointed to quality problems with certified seed distributed by the PMCs. We identified two main reasons for this. The first is inadequate inspection of growers' fields, including the existence of rent-seeking and collusion in the regulatory services. Key informants from the regulatory authority admitted that quality control is seldom carried out per the required standards because of the limited number and competence of field inspectors, insufficient cars for fieldwork, and inadequate facilities to conduct germination tests. Although inspectors denied this, seed producer farmers claimed that inspectors made unfair decisions based on bribes and that there was a lack

of transparency around field inspection decisions, quality approval, and distribution of certification tags. For instance, key informants from a commercial out-grower group complained that the seed they jointly produced on 15 hectares of land was rejected because inspectors found a smut contaminant in just one of the fields. They also mentioned that regulatory staff secretly distributed certification tags to some producers who had not undergone seed quality procedures. Consistent with the PSSDS, recent reports point to the lack of an independent regulatory authority as the main reason for poor seed quality control and certification services in Ethiopia [138,178,179], a situation that Tripp and Louwaars [180] argue can open the door to rent-seeking and collusion. The second reason is the lack of strict quality control at later stages in the value chain, particularly the work carried out by the public enterprises to collect seed from out-growers and clean and package it for distribution. Key informants indicated that it is not uncommon to combine seed from different agroecologies, as well as with seed leftover from the previous year, which explains why farmers in both districts complained of quality problems. To mitigate for this, there should be control along all parts of the value chain so that farmers can trust the seed that they are buying.

In terms of access to certified seed, the PSSDS notes that the price of certified seed in Ethiopia is relatively low compared to neighboring countries and thus does not consider this to be a major issue, beyond strengthening credit and savings schemes for resource-poor farmers, especially FHHs. This is generally true for commercial farmers in Heexosa, who explained that the costs of other inputs such as fertilizer and especially pesticides (given the heavy use) are more expensive than seed. Nonetheless, they considered the price unfair, claiming that the production gain from certified seed did not justify the cost. Key informants from the district bureau and research institutions felt that farmers did not understand all the costs implied in certified seed production. However, given shortcomings in quality control and the availability of less expensive, high quality seed from the intermediate sector, this also raises questions about the cost efficiency of certified seed production.

For resource-poor farmers, improving their purchasing power through credit and savings is one strategy to increase access to certified seed, especially for young farmers who are interested in using full-package technologies. However, until greater quantities of certified seed are available, it is likely that wealthier model farmers and out-growers will continue to have privileged access, thus increasing supply is key. Furthermore, our findings point to access problems even for seed of high-value varieties like Quncho obtained through social networks. Strategies to improve seed access should therefore extend beyond certified seed and include "orphan districts" like Gindabarat.

The formal seed system and related extension services and agricultural programs have provided opportunities for access to information and led to the impressive empowerment of FHHs in Heexosa. These findings show that progress in gender-responsive extension [181] is possible, but more efforts are needed to create opportunities for women in MHHs and to expand supports in marginal districts with few agricultural development actors.

### 6.3. Intermediate Seed System

The intermediate seed system has emerged during the past decade in Ethiopia as a way to increase farmers' access to seed and build local economies through decentralized community-based seed production and distribution. The ambition is to promote the development of independent, self-sustaining seed enterprises that address local needs and demands, especially for self-pollinating crops and specific agroecologies that are not met by the formal system. Given that it is relatively new, the interventions proposed by the PSSDS focus primarily on developing community-based seed production and marketing, including:

- Providing technical, financial, and infrastructure support for community-based seed producers to increase their capacity for QDS production and develop viable local seed businesses;

- Linking community-based seed producers to multiple marketing strategies and distribution channels, e.g., DSM agents and local market to facilitate access by farmers;
- Increasing community-based seed producers' access to diverse crop varieties for multiplication by linking them through contractual agreements with research institutes, the national genebank, and well-established CSBs.
- *Cross-linkages—intermediate, formal and informal:* leveraging social seed networks to increase distribution and access by farmers of all types of seeds (informal, QDS and certified); exchanging knowledge and skill among seed sector actors; formalizing promotion of all varieties (local, open pollinated and hybrid) based on farmers' needs through bureau of agriculture/government agricultural extension in collaboration with farmers organizations, NGOs, genebank and agricultural research.

Our findings show that community-based seed produced by CSBs and SPCs have made contributions to seed security in terms of availability, affordability, and quality. The growth and expansion of SPCs have mainly been supported by ISSD program in collaboration with government institutions in the seed sector. Through its 10-year intervention in Ethiopia, ISSD has established or strengthened 270 SPCs and mainstreamed the SPC approach to seed production and distribution in 50 government and development institutions, including the Ethiopian Agricultural Transformation Agency and German Federal Enterprise for International Cooperation (GIZ). Nationally, these SPCs have produced and distributed 392 varieties of 35 crop species [149]. This shows a growing positive contribution of SPCs to the availability and access to good quality seeds of diverse crop varieties [77,182].

Although the overall contribution to wheat and teff seed supply in the study districts remains limited, the most significant impact of ISSD is probably in terms of institutional innovation, especially with regard to the intermediate system. The program has successfully encouraged the government to facilitate SPCs' access to input and service providers (e.g., credit, source seeds, technical and management training) and infrastructure development [138] as well as to develop/adjust relevant policies and regulations, including the PSSDS itself as well as the QDS directive [149].

That said, implementation still lags behind. In general, SPCs in Heexosa are still mainly operating as out-growers for the formal seed system, with the sale of the 15% they retain as their main contribution to the intermediate seed supply system. While this has increased farmers' access to more affordable seed of improved varieties the volumes are still low. Further investments in terms of technical, financial, and infrastructure support are required to build their capacity to become independent seed enterprises that can meet local seed demand. Moreover, there is no QDS certification provided to the SPCs in Heexosa, as seed laboratories have prioritized seed certification in the formal system over QDS certification due to capacity constraints. Key informants and recent reports [139,179] indicate that there are gaps in the technical capacity of the SPCs' internal quality control committees, which QDS could help to address. Nonetheless, our findings indicate that farmers in the study districts are satisfied with the quality of community-based seed from SPCs and CSBs, consistent with recent evaluations of SPC seed supply in Ethiopia [183,184].

The PSSDS includes CSBs as relevant actors for commercial production and distribution of local varieties. To date, CSBs in Ethiopia have played a more important role in terms of making a diversity of locally adapted varieties available to farmers through low-interest seed loans for members. We are not aware of any that have integrated commercial seed production and marketing into their operations, as has been done by CSBs in other countries like Nepal [185]. Given the CSBs' experience with local varieties such as durum wheat, they could play a role in QDS seed production and marketing of improved durum wheat varieties that are currently on the shelf. There are still relatively few (approximately 30) CSBs in Ethiopia [186], despite its promotion since the mid-1990s [187], thus the overall contribution to seed supply remains limited.

From the perspective of pluralistic seed systems, a diversity of seed sources provides smallholder farmers a greater choice of seeds and varieties [17,188,189]. SPCs and CSBs

are both contributing to increasing production and distribution of certified, QDS and local seed. Our findings also suggest that there may be an opportunity to scale out impacts of the intermediate sector, particularly in marginal areas, by linking CSBs and SPCs to existing community institutions like Afoosha, that have a strong local governance based on principles such as trustworthiness and altruism [153]. Overall, we find that the intermediate system represents a huge potential to foster linkages between formal and informal systems and increase the availability and access of diverse seeds and varieties to farmers. To meet this potential, government needs not only to invest in expanding existing programs and capacities, but also to resolve conflicts between conservation and agricultural research and development institutions.

## 7. Conclusions

This study contributes to the seed security and seed system literature by revealing some of the social, political, and institutional constraints and opportunities that underlie chronic seed insecurity among smallholder farmers in Ethiopia. While the seed security literature has focused on post-disaster settings, our findings from a "normal" growing season reveal evidence of seed insecurity in all four dimensions (varietal suitability, availability, access, and quality) for both the subsistence and commercially oriented production systems examined.

In broad terms, a number of seed security challenges are common to both subsistence and commercially oriented systems, such as seed quality issues relating to lack of varietal purity and storage of own-saved seeds and the need for new varieties to adapt to diseases. However, the nature and severity of the challenges differ particularly as they relate to the formal seed system. For example, although farmers in both districts suffer from insufficient availability and access to seed from the formal system, this is more marked in the subsistence-oriented district where crop improvement research and formal seed supply channels are nearly entirely absent. On the other hand, in the commercially oriented production system, there is a lack of availability of certified seeds and a lack of access to farmer preferred traditional varieties. Furthermore, our findings indicate that the heavy presence of seed sector actors in the commercially oriented district has led to differences in seed access between socio-economic groups. It seems the targeting of female headed households by the extension services have indeed increased this group's access to certified seeds. Another group with better access to certified seeds are wealthy farmers aligned with the government who are favored for positions as model farmers and out-growers. Our study further shows that high grain/seed price constrains access not only for seed from the formal seed system but also for high value seed/grain accessed through social networks. Overall, we conclude that farmers are navigating between an eroding traditional system and a dysfunctional formal system.

Our analysis of the PSSDS shows a good alignment between the policy's proposed priority interventions and farmers' seed security challenges. In large part, this is due to the pluralistic approach taken in the policy that puts farmers at the center of seed sector development by promoting complementarity between value-chain components of each seed system and integrating their activities, in contrast to the dominant linear model to seed sector development in developing countries. However, our field-based findings show that the operationalization of the policy lags behind, with investments in the informal seed system largely missing from government programs, whereas the main source of proposed funding for this system (i.e., access and benefit-sharing funds) is unlikely to materialize. Some improvements have been made in the formal system, and the overall integration of all seed systems for holistic seed sector development, however progress is hampered by vested political, organizational, and economic interests within key seed sector institutions, as well as insufficient resources and capacity. The intermediate "community-based" seed system shows promise, though limited in scale, and in the subsistence-oriented production system, restricted only to investments by NGOs. More generally, the implementation of

the PSSDS faces fundamental problems with key actors not adapting their mandates and programs to reflect the PSSDS' pluralistic approach.

Overall, our study suggests that pluralistic seed system development can provide a path to seed security in developing countries. This requires that well-designed policies like the PSSDS lead to investment at scale to strengthen the informal seed system and dysfunctions in the formal system, while investing in the intermediate system. However, for this to happen, historical, institutional, political, and social factors that underlie the current (dys)functioning of the seed sector need to be understood and tackled. Context specific research that examines this complex interplay of factors is crucial. Finally, the potential that the intermediate seed system shows call for more investment, but while some improvements have been made in the formal system, vested political, organizational, and economic interests within key institutions represent major obstacles that must be overcome to achieve truly integrative and inclusive seed sector development.

**Author Contributions:** Questionnaire preparation, T.H.M.; Tool pre-testing, T.H.M.; Conceptualization, T.H.M., O.T.W. and S.P.D.; Methodology, T.H.M. and O.T.W.; Data collection process, T.H.M.; Data cleaning and analysis, C.M. and T.H.M.; Draft preparation, T.H.M.; Draft review editing, O.T.W., S.P.D., R.H. and T.H.M.; Supervision, O.T.W.; Funding acquisition T.H.M. and O.T.W. All authors have read and agreed to the published version of the manuscript.

**Funding:** This research was funded by RESEARCH COUNCIL OF NORWAY, grant number RCN 277452 and RCN 4850060008.

**Institutional Review Board Statement:** The study was conducted according to the guidelines for research ethics of the Norwegian University of Life Sciences and Norway's guidelines for research ethics in the social sciences, humanities, law, and theology. Based on these guidelines, we submitted a notification form to the Norwegian Center for Research Data (NSD) prior to commencing data collection and received approval from NSD (Harald Hårfagres gate 29, N-5007 Bergen, Norway) on 18.09.2017. The guidelines did not require an explicit ethics approval in Ethiopia. We informed the state, district and village authorities and they gave us permission to carry out the surveys, interviews, and discussions.

**Informed Consent Statement:** Informed consent was obtained from all subjects involved in the study.

**Data Availability Statement:** The data presented in this study are available from the corresponding author upon request. The data are not publicly available due to protection of privacy linked to research participants' personal data.

**Acknowledgments:** This paper is based on research project conducted in completion of the PhD degree by the first author, under supervision of the last two authors. We are very grateful to the Oromia Bureau of Agriculture and Natural Resources and the district Bureau of Agriculture and Natural Resources in Gindabarat and Heexosa for facilitating the household surveys and FGDs. We appreciate the generous participation of farmers, public, non-government and private seed sector actors involved in Ethiopian seed systems for discussions and interviews. We thank Tola Gemechu Ango (Stockholm University) for making the map and Habtamu Alem (Norwegian Institute of Bioeconomy Research) for valuable advice on data analysis. We also thank Åsmund Bjørnstad and Trygve Berg in Norwegian University of Life Sciences (NMBU) for providing important inputs to earlier versions of this paper. THM wishes to thank the NMBU for the scholarship to carry out this research and the Research Council of Norway (RCN) for financial support for his fieldwork under grant number (RCN 277452), and the International Maize and Wheat Improvement Center (CIMMYT) in Addis Ababa for hosting him and facilitating his fieldwork during the period. OTW, CM and RH are grateful for the financial support through the RCN funded project ACCESS (RCN 4850060008).

**Conflicts of Interest:** The authors declare no conflict of interest.

# Appendix A

**Table A1.** Roles of seed sector actors in contributing (↑) and/or constraining (↓) smallholder farmers' seed security of the teff-centered and subsistence focused farming system in Gindabarat district, and the wheat-centered and commercially oriented farming system in Heexosa district. Roles that have not yet had the intended effect are denoted with ↔.

| Actors (Gindabarat) | Seed Security Features | | | | |
|---|---|---|---|---|---|
| | Varietal Suitability and/or Adaptability [1] | Availability [2] | Access [3] | Quality [4] | Gender, Socioeconomic Status, and Age [5] |
| **Local government decision makers/experts** *District bureau of agriculture* | ↑ Bring pre-basic/basic seeds or early generation seeds (EGS) of improved varieties from agricultural research located in similar agroecology and conduct participatory variety trials together with farmers under different input packages and agronomic practices at FTC. ↑ Recognize and support participatory variety selection (PVS) of traditional varieties conducted by the community seed bank (CSB) group (see below) ↔ Requested support from regional government for variety testing and research on new technologies (not yet obtained) | ↑ Assess farmers' seed demand and determine quantity of certified seeds required ↑ Provide external support (e.g., administrative and financial management) for formally organized farmers for seed production (see below) in collaboration with District Cooperative Promotion Bureau ↔ Requested support from regional government for seed production and distribution to increase supply of improved seeds (not yet obtained) ↓ Did not establish seed reserve for seed system resilience in cases of disaster. ↓ Not aware about or did not request the Regional Bureau of Agriculture to provide certificate of competence for interested seed agents [8] and cooperatives for direct marketing of certified seeds to farmers and effective distribution | ↑ Determine share of certified seeds for peasant associations, enforce government prices, and support Primary Multipurpose Cooperatives (PMCs) during seed distribution ↑ Conduct field demonstration of new varieties at Farmer Training Centers (FTC) to increase awareness among farmers ↓ Despite weak evidence, the extension often promotes improved varieties as better yielding than traditional varieties ↔ Submitted requests for budget from regional government to build physical infrastructure (e.g., access road) to improve access to agricultural inputs and marketing outputs (not yet obtained) | ↑ Collect data from farmers and report events of poor performance due to low seed quality of certified seeds to regional bureau of agriculture to enforce commercial guarantee [6] and settle disputes ↑ Recognize the seed quality criteria that most farmers use [7] ↑ Support trainings on quality seed production and storage for members of an NGO-supported community seed bank (CSB) group (see below) ↓ No trainings provided on seed production and storage provided for individual households ↓ No technical training and infrastructure support for PMCs to increase their capacity to properly store seeds they receive from public seed enterprises | ↓ Extension services, technology promotion, and agronomic trainings prioritize model farmers (often the majority are male household heads), which marginalize women and youth ↔ Established women/youth leagues/federations at the local and district level to increase participation in agriculture development issues, but the structure is mostly utilized for political governance of the district by the leading party |
| **National/regional research** *Holeta Agricultural Research Center (HARC) and Debre Zeit Agricultural Research Center (DZARC)* | ↔ Send limited EGS samples of new varieties to the district agriculture bureau for use in participatory trials (see above), but most of them failed to adapt to the local environment [9] ↓ Do not conduct variety development and adaptation specific to the district agroecology | ↓ Do not produce and distribute early generation seed in the district because commercial seed producers are not present | ↓ Do not provide extension and training for DAs and lead farmers to increase awareness on varietal information and agronomic practices | ↓ Germination failure of seeds for PVS trials, due to delays in shipments/long shelf life | |

Table A1. *Cont.*

| Actors (Gindabarat) | Seed Security Features | | | | |
|---|---|---|---|---|---|
| | Varietal Suitability and/or Adaptability [1] | Availability [2] | Access [3] | Quality [4] | Gender, Socioeconomic Status, and Age [5] |
| **Local traders/markets and seed agents/agrodealers** *Traders of grain/seeds including farmers who sell at local markets* | ↓ Vendors/traders combine grains from different agroecological areas (lack traceability of source); this sometimes causes crop failure for teff if planted in the wrong agroecology | ↑ Bring diversity of grain from different areas to local marketplaces that farmers buy for food grain or seed ↓ Lack of local agrodealers hinders availability of improved seeds | ↑ Seeds sold at local markets are easily accessed (close by and timely available) ↓ Wheat seed sold or lent by traders/venders is often expensive due to low availability | ↓ Grain/seed sold at local markets generally rated by farmers as poor in terms of germination and purity | ↑ Local markets provide poor farmers (e.g., landless youth) access to grain/seeds when they cannot save seeds or consume their saved seeds. This is a last resort, due to poor quality of seed. |
| **Specialized seed producers and farmer organizations/groups** *Community Seed Bank (CSB) group, Primary Multipurpose Cooperatives (PMC) and Afoosha* [10] | ↑ The CSB group conducts PVS on pools of varieties from the local area, genebank, and other communities to identify varieties suitable for low input farms ↔ Occasionally, the PMCs distribute varieties that are not recommended for the specific local agroecology (e.g., hybrid maize for highland is sold to midland areas) | ↑ One CSB group produces limited quantities of local wheat and teff seeds ↔ The PMCs obtain certified seeds from the Ambo Farmers Union, but these often arrive too late and in insufficient quantities ↑ Afoosha maintain grain reserves for local food and seed relief ↓ There are no organized seed producers for improved varieties | ↑ The CSB group distributes seeds through a loan system with low interest repaid at harvest (10% in kind/seed) ↑ The PMCs sell certified seeds to users at government price ↑ Afoosha give free seeds to families affected by death or natural calamities | ↑ Farmers have positive perception of local seeds produced and communally certified by CSB group ↓ Farmers complain about poor quality of certified seed distributed by PMCs (e.g., hybrid maize and Quncho seeds) | ↑ CSB groups and Afoosha offer seeds to poor farmers and households affected by calamities (e.g., widows) ↑ Gender balance in the CSB group allows consideration of women's priorities in seed multiplication (e.g., local barley varieties that were introduced from other areas) ↓ Female household heads have limited access to certified seeds from PMCs that are dominated by men |
| **Non-governmental Organizations/ Development agencies/Inter-governmental organizations** *Movement for Ecological Learning and Community Action (MELCA-Ethiopia)* [11] | ↑ MELCA trains men and women CSB members on PVS of local varieties to meet diverse environmental and socioeconomic needs | ↑ MELCA brings seed/germplasm from the national genebank and other communities for multiplication to increase availability of traditional seeds | ↑ MELCA supports seed loan system managed by CSB group (see above) ↓ MELCA's training crop diversification often promote traditional varieties as better varieties than improved varieties | ↑ MELCA supports communal seed certification through CSB's seed committee ↑ MELCA supported construction of community seed bank facility for improved seed storage | ↑ MELCA supports CSB groups in organizational capacity building including administration, seed, and financial management through balanced representation of different farmer categories (gender, age, and wealth categories) |

Table A1. *Cont.*

| Actors (Gindabarat) | Seed Security Features | | | | |
|---|---|---|---|---|---|
| | Varietal Suitability and/or Adaptability [1] | Availability [2] | Access [3] | Quality [4] | Gender, Socioeconomic Status, and Age [5] |
| **Smallholders** *Own seed production and social networks* | ↑ Farmers verify the varietal suitability of seed provided through social networks (neighbor certification) ↓ Farmers lack sources of new varieties to adapt to declining soil fertility and increasing rust for wheat production | ↑ Most farmers produce and save own wheat and teff seeds ↑ Lead farmers save seeds from adaptation trials if they prefer a variety and multiply for their own use and exchange with other farmers ↓ Most landless and poor households do not save enough seeds to meet their needs | ↑ Farmers loan (i.e., with interest), sell, or exchange seeds with friends, neighbors, or family ↑ Better-off individuals provide cash loans that are used for seed purchase ↓ The custom of seed gift is abandoned | ↑ Farmers perceive quality of own seeds as good ↑ Farmers maintain varietal purity of high-yielding improved teff through appropriate selection and seed handling ↓ Occasionally untimely rain combined with lack of good storage facility cause damages in household seed stocks | ↔ Social networks and moneylenders help landless and poor households to access seeds on credit, but interest rates are high, making repayment difficult. ↑ Younger farmers often access an improved teff variety through social networks to increase productivity on small landholdings ↓ Most lead farmers are men, limiting women's access to new varieties |
| **Others not active in Gindabarat** | The following seed sector actors are not active in Gindabarat: Regulatory bodies (Ambo seed quality control and certification laboratory of the Oromia Agricultural input regulatory authority); International research (e.g., CIMMYT, ISSD); Public/private seed sector (Ethiopian Seed Enterprise/ESE, Oromia Seed Enterprise/OSE and commercial private farms); and Private sector processors (e.g., private small-scale milling) | | | | |
| Actors (Heexosa) | Varietal Suitability and/or Adaptability | Availability | Access | Quality | Gender, Socio-Economic Status and Age |
| **Local government decision makers/experts** *District bureau of agriculture* | ↑ Conduct participatory variety adaptation trials of new varieties together with farmers under different input packages and agronomic practices at FTC ↑ Recognize and support PVS of traditional varieties in marginal areas (e.g., higher elevations) | ↑ Support market-led seed supply to increase availability of certified seeds and locally produced quality declared seeds (QDS) ↓ Supported seed agents and cooperatives to get certificate of competence from the Regional Bureau of Agriculture for direct seed marketing of certified seeds to farmers and effective distribution ↑ Collect demand from farmers and determine quantity of required certified seeds ↑ Support CSB seed production to increase seed supply through farmer training ↓ Did not establish seed reserve for seed system resilience in cases of disaster | ↑ Determine share of certified seeds for peasant associations, enforce government prices, and support PMCs during seed distribution ↑ Conduct field demonstration and seed fairs (field days) to increase awareness and information on new seed varieties and their characteristics ↓ Despite weak evidence, the extension often promotes improved varieties as superior varieties for yield and disease resistance and discourage use of traditional varieties | ↑ Monitor farmers involved in the production of certified seeds and seeds for the CSB for implementation of good agronomic practices [12] ↑ Collect data from farmers and report events of poor performance due to low seed quality of certified seeds to regional bureau of agriculture to enforce commercial guarantee[13] and settle disputes | ↔ Support women's participation in seed producer cooperatives and trainings, but limited to women household heads ↑ Encouraged and recruited women household heads as model farmers |

**Table A1.** *Cont.*

| Actors (Heexosa) | Varietal Suitability and/or Adaptability | Availability | Access | Quality | Gender, Socio-Economic Status and Age |
|---|---|---|---|---|---|
| **Regulatory bodies** *Asella seed quality control and certification laboratory of the Oromia Agricultural input regulatory authority* | | ↓ Strict certification and rejection of seeds produced by contract cluster groups and individual farmers reduced availability of certified seeds to some extent, but limited sales inspection allowed seed sellers to supply rejected seeds though sometimes adulterated | | ↓ Inadequate human resources to conduct field inspection at all seed production stages and limited laboratory facilities and testing protocols to conduct quality tests of all seeds from producers' plots contributing to ineffective seed certification ↑ Provide training for organized producers on quality seed production and management | ↑ Provide technical training on seed production, processing, and storage for internal seed quality control committee of seed producers, including female members |
| **National/regional research** *Kulumsa agricultural research center (KARC) and Asella Agricultural Engineering Research Center (AAERC)* | ↑ Since its establishment, KARC has produced about 70 wheat varieties [2] with different merits and conducted adaptation trials in collaboration with agriculture bureau at FTCs and on farmers' plots to ensure suitability to farmers' environmental and socioeconomic conditions ↓ Disease-resistant wheat varieties are generally lacking, and production is impossible without pesticides ↓ Variety replacement rate is low due to slow release of new varieties and low seed multiplication of released varieties | ↑ KARC produces EGS and makes these available for public seed enterprises, unions, and SPCs ↓ However, not enough quantity EGS are produced and made available for the multiplication of successive generations of seeds (e.g., certified seeds) by seed producers | ↑ KARC supports field demonstration and extension to increase awareness of farmers and development agents on varietal information and good agronomic practices ↓ Lack of strong unit in agricultural research is the cause for weak coordination for sustainable EGS access and supply and loose responsibility of EGS multiplication | ↑ KARC conducts internal quality control of its EGS before distribution for adaptation trial and multiplication ↓ Poor quality of EGS is sometimes delivered due to limited human resources, equipment, and infrastructure ↑ AAERC provides training in pre-harvest, harvest, and post-harvest technologies (e.g., cleaning combines to avoid varietal mixture) | ↑ KARC involves some female household heads in variety testing and adaptation trials |
| **International research** *CIMMYT* | ↑ CIMMYT brings advanced lines of wheat seed samples from other countries for the testing and identification of adaptable variety ↑ Together with KARC, CIMMYT develops disease-resistant wheat varieties ↓ CIMMYT does not work on teff | ↑ CIMMYT provides support to KARC for the multiplication of large quantity of EGS | ↑ CIMMYT organizes exposure visits for farmers, development agents, and entrepreneurs to increase awareness about new varieties | ↑ CIMMYT ensures the seed samples it imports are free from quarantine pests | ↑ CIMMYT provides training of trainers and researchers on gender issues for mainstreaming in crop improvement research |

**Table A1.** *Cont.*

| Actors (Heexosa) | Varietal Suitability and/or Adaptability | Availability | Access | Quality | Gender, Socio-Economic Status and Age |
|---|---|---|---|---|---|
| **Local traders/markets** *Traders of grain/seeds including farmers who sell at local markets* | ↑ Recycled wheat variety from midland areas is perceived by farmers to have better yield and disease resistance in highland agroecological conditions and vice versa | ↑ Bring large quantities of grain/recycled or traditional seeds from all agroecological areas and make these available at local markets | ↑ Grain/seed sold by traders/vendors is easily accessible (nearby) ↓ Seed sold or lent by traders/vendors is expensive (especially teff) | ↓ Seed purchased from traders/venders is not quality controlled and generally perceived by farmers as having poor quality | ↑ Local markets provide poor farmers access to grain/seeds when they cannot save seeds or consume their saved seeds. This is a last resort, due to poor quality of seed. |
| **Public/private seed sector** *Ethiopian Seed Enterprise/ESE, Oromia Seed Enterprise/OSE and Seed agent/agrodealers* | ↓ Sometimes, wrong varieties are distributed in wrong agroecological areas | ↔ The seed enterprises produce and supply most of certified seeds via government-controlled distribution channels, but quantities are insufficient (especially teff) and distribution is often delayed ↔ The seed enterprises also produce and supply EGS to other seed producers, but quantities are insufficient ↑ Recent increase in number of seed agents improved availability of certified seeds in wider coverage of agro-ecologies ↑ Prioritize seed supply to severely seed insecure areas when disaster hits | ↓ High price discourages farmers from using certified seeds ↑ Recent contract-based direct seed marketing (DSM) through seed agents has increased timely supply within easy reach, but the agents sometimes increase the price against the agreement and make it unaffordable for the poor ↓ The involvement of the private sector that sells seed is generally limited ↓ EOSAs often promote traditional varieties as better varieties than improved varieties | ↔ Supply certified seeds but sometimes quality fails to meet the required standards, especially for carryover seeds ↑ Train contract cluster groups and members of seed producer cooperatives in quality seed production and management as well as agronomic practices in wheat production ↓ Seed agents lack good storage facility for temporary stocking until they sell seeds or return leftover seeds, which sometimes affect quality | |
| **Specialized seed producers and farmer organizations/groups** *Seed Producer Cooperatives (SPCs), Community Seed Bank (CSB) groups and Individual out-growers* | ↑ SPCs produce seeds of many preferred and adapted crops and varieties (e.g., self-pollinated, high-yielding, and marketable cereals and legumes) that were not easily available through the public seed enterprises in the past ↑ CSB groups conduct PVS and produce seeds of locally preferred varieties for low-input farms (especially in high-elevation areas) ↑ SPCs and CSBs are in the center of the farmers' village and know their customers in terms of varietal suitability to the agroecology and availability and affordable price | ↑ SPCs and individual out-growers produce large quantities of seeds locally or within easy reach ↔ PMCs receive seeds from Heexosa Farmers Union but in insufficient quantities ↓ No organized group of farmers produce teff seeds | ↑ Seed producer farmers/out-growers can keep enough seeds (up to 15%) for own use ↑ Seed price is lower than the prices of public/private companies ↑ SPCs are in the center of the farmers' village and set seed prices that their customers can afford ↑ CSB gives seed loan that is paid with low interest (10% in kind/seed) | ↑ Farmers have positive perception of seeds produced by SPCs and communally certified by internal seed quality control Committee of SPCs and CSB ↓ SPCs lack a seed cleaner machine, mini seed laboratory equipment such as moisture testers, and germination Petri dishes for seed quality checks ↓ Experts see farmers' confidence in their long agriculture experience as a guarantee for their capacity to control seed quality, instead of using skilled personnel and establishing laboratory facility, as the cause for sporadic poor seed quality produced by SPCs | ↑ Cooperatives support to farmers in provision of basic seed, training, and supervision through linking farmers with research institutions and input, and service providers emphasize women participation ↓ However, the number of women members in SPCs is very low |

**Table A1.** *Cont.*

| Actors (Heexosa) | Varietal Suitability and/or Adaptability | Availability | Access | Quality | Gender, Socio-Economic Status and Age |
|---|---|---|---|---|---|
| **Non-governmental Organizations/ Development agencies/Inter-governmental organizations** *USAID, FAO, Hunde Oromia and Ethio-organic seed action (EOSA) and ISSD Programme* | ↑ EOSA brings seed/germplasm from other communities and the national genebank and conducts PVS of local and improved varieties to meet diverse environmental and socioeconomic needs in marginal areas ↑ ISSD introduced an innovative approach called crowdsourcing and participatory variety selection that aims to outsource multiple improved and farmers' preferred varieties of different crops to many volunteer farmers who are willing to grow and share the selected variety in their locality | ↑ FAO, USAID, and Hunde provide seed aid when disaster hits and support seed multiplication ↑ EOSA supports CSB group to multiply traditional seeds/varieties selected through PVS ↑ ISSD provides financial and technical support to agricultural research (mainly regional) and OSE in contract-based multiplication of a large quantity of EGS ↑ ISSD provides financial, technical, and administrative support to increase the number and capacity of SPCs and seed agents for the production and distribution of large quantities of self-pollinating crop varieties that are neglected by public seed enterprises and private companies | ↑ USAID, FAO, and Hunde provide vouchers to assist resource-poor households to access seeds according to their needs ↑ EOSA supports the CSB group in administering seed loans (see above) ↑ ISSD supports linkage between SPCs and financial institutions for credit as well as EGS sourcing institutions to increase SPC's access to pre-basic and basic seeds ↑ ISSD promotes small seed pack sizes based on the average land size that smallholders cultivate for each crop to increase access to required quantities of seeds at affordable prices | ↑ FAO, USAID, and Hunde distribute certified, and quality declared seeds ↑ EOSA supports communal seed certification through seed farmer committee ↑ EOSA trains CSB members on crop diversification, good quality seed production, and storage ↑ ISSD supports training of SPC members on clustering, isolation, field management, and roughing to remove off types as well as seed value addition (cleaning, grading, treating, packaging, and labeling) to increase quality through technical training, exchange visits, resource mobilization, and linking them with service providers (e.g., credit institution for purchase of processing machines and seed labs for coaching) | ↑ EOSA also trains farmers in organizational governance and women participation to ensure sustainability ↑ ISSD promotes gender-sensitive crop and varietal preference for deployment in its crowdsourcing and PVS activities |
| **Private sector processors** *Heexosa Multipurpose Union and private small-scale milling factories* | ↓ Sometimes, the Union distributes certain varieties to areas for which there is no demand ↓ Sometimes, the Union distributes seeds to the wrong agroecologies | ↑ The Union procures certified seeds from SPCs, ESE, OSE, and private seed companies for distribution through its PMCs ↓ PMCs do not participate in seed demand assessment and depend on unrealistic data collected by extension agents and wrong quota allocation, which restricts seed supply/availability | ↑ The Union collects seeds and transports to selling points ↑ The Union and private small-scale milling factory purchases grains for milling at a reasonable price from primary cooperatives, allowing farmers to get income to purchase seeds for the upcoming planting season | ↓ Sometimes, the Union distributes untraceable poor-quality seeds (including carryover seeds without laboratory seed tests) due to lack of accountability and transparency in the conventional seed distribution system | ↑ The Union trains cooperative members including women and youth on business management ↑ Provides benefit for male and female household heads through agro-commodities procurement |

**Table A1.** *Cont.*

| Actors (Heexosa) | Varietal Suitability and/or Adaptability | Availability | Access | Quality | Gender, Socio-Economic Status and Age |
|---|---|---|---|---|---|
| **Smallholder farmers** *Own seed production and social networks* | ↑ Own produced seed of recycled/traditional varieties is comparable in productivity compared with certified seeds ↓ Farmers lack new disease-resistant wheat varieties ↓ Low productivity/high labor demand of teff varieties causes most farmers to abandon its cultivation | ↑ Most farmers produce and save own seeds ↑ Farmers save seeds from adaptation trials organized by district bureau of agriculture or CIMMYT for own use ↓ Some poor and landless households do not save enough seeds to meet their needs ↓ As secondary crop, few farmers grow teff, and its seed is not available in many villages | ↑ Seeds sold by trusted farmers (e.g., neighbors) are close and affordable ↑ Farmers loan (i.e., with interest), sell, or exchange seeds with friends, neighbors, or family or provide cash loans for seed purchase ↑ Rich farmers often access new seeds, and they multiply and sell their produce as seeds to other farmers ↓ The custom of seed gift is absent | ↑ Farmers perceive quality of own seeds, and those purchased/exchanged from fellow farmers, as good (known quality, neighbor certification) ↑ Most farmers use pesticides and recommended polypropylene bags to store wheat seed for one season | Farmers see high seed insecurity among landless and poor households in lowland areas Farmers who do not save their own seeds mostly depend on local exchange or purchase recycled improved seeds locally |

[1] Varietal traits meet farmers' preferences including adaptation to local environment and production conditions, market demand, culinary and cultural needs, livestock feed, construction, and soil fertilization. [2] Physical existence of desired seeds in enough quantity in a reasonable proximity (spatial availability) for critical sowing periods (temporal availability). [3] Means to acquire seeds such as cash, credit, social network, and transportation with affordability and awareness/information. [4] Seed is healthy (free from disease/pest), has good physical qualities (not broken/cracked/shriveled), has good genetic and physiological qualities (good germination, optimum moisture content, genetic purity, and vigor), free from weeds and poses preferred color/size/shape/taste. [5] Impacts on seed security by gender, socioeconomic status, and age (cross-cutting). [6] Commercial guarantee: obtained from purchased seed usually bought locally from known seed dealer and oral, commercial, and often legal assurance is given. [7] Known quality: obtained from on-farm saved seed; 'neighbor certification': obtained from seed saved by family members and neighbors on trust. [8] A seed agent is an individual or institutions who sell seed to farmers on behalf of seed producer/s. They should acquire certificate of competence (CoC) from the government based on the requirements in the 2018 COC directive No.2/2010. The seeds are sold at a fixed producer price and commission is based on the amount sold. [9] In the last decade, only three teff varieties (Quncho, Kora and Guduru) and two wheat varieties (Digelu and Hidase) have been adapted to the environment and are liked by farmers. [10] Afoosha is an indigenous local social institution established in most communities in Ethiopia to provide financial and other types of support when a family member dies. In Gindabarat, we found that Afoosha groups have established grain reserves in most peasant associations to support poor families affected by calamities; these reserves are increasingly used as seed by those affected. [11] MELCA-Ethiopia is a local NGO supported by the Development Fund of Norway. [12] Soil use and fertility management (e.g., fertilizer application), crop rotation, row planting, recommended distance between plots per species, and crop protection. [13] Commercial guarantee: obtained from purchased seed usually bought locally from known seed dealer, and oral, commercial, and often legal assurance is given.

**Table A2.** Analysis of the Ethiopian pluralistic seed system development strategy [3] for improving the functioning of seed systems and smallholder farmers' seed security. The table illustrates the policy interventions identified and recommended for implementation based on analysis of issues and constraints that form systematic bottlenecks across the seed value chain in all the three seed systems (informal, formal and intermediate). The interventions presented here cover all levels of seed system governance with relevance to the availability, access, quality and varietal suitability of seeds that are required to meet farmers' needs.

| Seed System | Seed Security Features | | | |
| --- | --- | --- | --- | --- |
| | Varietal Suitability [1] | Availability [2] | Access [3] | Quality [4] |
| **Informal** | • Improve linkage between farmers and agricultural research (e.g., institutionalize PVS/PPB in crop improvement, identify and train seed selectors, and involve them in multiple stage of conventional variety breeding) especially by ensuring participation of women who can provide feedback on characteristics beyond yield, e.g., health-related issues and traits for food preparation <br> • Strengthen pre-breeding component of the national genebank and promote increased access to germplasm for PVS, PPB and conventional breeding programs of the national agricultural research centres. | • Promote application of appropriate agronomic practices (recommended seed and fertilizer rate, pest and weed management, crop rotation, intercropping, etc.) to enhance yield and quantity of seeds produced by farmers <br> • Support ex situ and in situ linkages to strengthen management of local genetic diversity through CSBs and strengthen their capacity to multiply seeds of diverse local varieties and reduce risk of genetic erosion <br> • Set up an efficient *National Seed Emergency System* to effectively respond to natural/manmade disasters and increase seed security, e.g., develop a national strategy for seed reserve and emergency assistance (setting aside adequate local seed reserve), set up an independent institution for seed security assessment, planning and implementing seed assistance activities. | • Strengthen CSBs' capacity to facilitate farmers' access to seed by providing revolving funds and designing a system where CSBs receive financing from Ethiopia's access and benefit sharing agreements <br> • Set aside an independent fund (revolving fund) that is specifically dedicated for emergency seed aid and coordinating seed aid actors' interventions to increase access to seeds by affected farmers <br> • Strengthen and promote innovative local seed marketing networks for efficient seed diffusion (e.g., promote field days, community seed fairs, open markets and CSBs) | • Strengthen the technical and infrastructural capacity of CSBs for improved seed storage <br> • Develop a special seed quality control system for emergency seeds, i.e., quality/quarantine checks for insect, pests, plant diseases and noxious weeds <br> • Strengthen farmers' awareness in proper seed management methods and improve access to affordable implements, e.g., use improve seed processing techniques (harvesting at maturity, keep physical and varietal purity and use post-harvest technologies) and effective seed storage techniques (e.g., proper drying and use of hermetic bags) <br> • Disseminate best practices in seed/varietal selection and maintenance for the informal system, e.g., developing manuals on proper seed/varietal selection (including farmers' criteria and scientific techniques for local varieties) and recycling practices or quality maintenance techniques (e.g., varietal purity for improved varieties) and well as using demonstration plots at Farmer Training Centers for raising awareness on varietal traits and agronomic practices for good quality seed production |

**Table A2.** *Cont.*

| Seed System | Seed Security Features | | | |
|---|---|---|---|---|
| | **Varietal Suitability** [1] | **Availability** [2] | **Access** [3] | **Quality** [4] |
| **Formal** | • Strengthen linkage between federal and regional research centers and increase their financial viability to embrace participatory variety development to incorporate traits beyond yield, involving especially women<br>• Establish a federal regulatory authority independent from variety development entities; build its capacity (e.g., finance, human resource) for varietal evaluation, release, registration and plant variety protection through development or amendment of laws, regulations and directives; support technical capacity building and implementation of national seed proclamations and regulations at regional level | • Support training of staff/certified agents in seed demand estimation and local market assessment to improve planning of early generation seed (EGS) and certified seed production<br>• Delineate and enforce roles and responsibilities among actors to avoid overlap and increase production (e.g., designate production of breeder seeds to research institutes; pre-basic and basic seeds to ESE; certified seeds of self-pollinating food crops to ESE and regional public seed enterprises; certified seeds of commercially attractive crops such as hybrid maize and vegetable seeds to international and national private seed companies)<br>• Expand certified seed production using contractual out-grower scheme between public/private seed companies and SPCs/cluster groups; build capacity of out-growers and provide incentives (subsidized production inputs, credit, seed cleaning/processing services, transportation, land, improved business planning, marketing, and operations management) | • Promote direct seed marketing (not involving the bureau of agriculture and PMCs) to overcome delays that occur in the government's centralized conventional seed supply system and increase timely access by farmers<br>• Implement open pricing mechanism for seed producers of public varieties and eliminate government price-setting by the Ministry of Agriculture and corresponding bureaus at regional levels to hasten timely access to quality seeds<br>• Provide financial services to farmers to increase input affordability, with emphasis on female-headed households (with target to increase their access to agricultural credit from 5.4% baseline to 30% of female-headed households. | • Establish more robust transportation, logistics, and seed storage systems, and better financial and technical support for seed distributing agents to avoid seed adulteration or mixture with grain/infested by pest/physically damaged during storage and transportation<br>• Ensure accreditation of seed laboratories and issue certificate of competence for multiregional enterprise at federal level based on set of criteria for quality seed production and supply<br>• Build capacity of seed laboratories (e.g., training of technical lab and field staff, equip lab facilities, vehicle access, sufficient budget) for proper coordination and planning of controls with seed producers, e.g., field inspection, seed quality testing, decision on rejections/approval of seed fields and seed lots<br>• Build capacity of agricultural research to ensure maintenance of breeder seeds |

**Table A2.** *Cont.*

| Seed System | Seed Security Features | | | |
|---|---|---|---|---|
| | **Varietal Suitability** [1] | **Availability** [2] | **Access** [3] | **Quality** [4] |
| **Intermediate** | • Focus CBSP seed production on less profitable crops such as local and improved varieties of self-pollinating crops (developed through conventional plant breeding, participatory variety selection and participatory plant breeding) with superior traits to fulfill seed production needs unmet by the formal system<br>• Increase CBSPs' access to diverse crop varieties for multiplication by effectively linking them through contractual agreements with research institutes, the national genebank, and established CSBs | • Establish a revolving fund and/or provide credit guarantee to lending institutions to increase access to finance for CBSPs for capital and production investment as well as building technical and infrastructure capacity that will eventually contribute to expansion of Quality Declared Seed (QDS) production<br>• Improve operational efficiency and sustainability of existing CBSPs to transition them into independent business entities that produce large volumes of QDS, i.e., by providing them targeted and continuous training, strengthen their logistic and storage capacity, promote proper clustering of plots and fair pricing mechanisms for out-growers to maximize seed recovery rate<br>• Promote direct contractual linkage between CBSPs and public seed enterprises instead of quota scheme through bureau of agriculture to increase access to EGS, thus increasing production of QDS<br>• Increase women's participation in SPCs, provide skill development support for women in seed production and infrastructure to increase their capacity for seed production. | • Link CBSPs to multiple marketing strategies and distribution channels (direct seed marketing agents, local market, etc.) to facilitate access by farmers | • Replace conventional quality standards that are too stringent for CBSPs with QDS regulatory system to ensure basic seed quality<br>• Build internal capacity of CBSPs for quality seed production through agronomic training, infrastructure development and quality input supply (e.g., EGS) |

[1] Varietal traits meet farmers' preferences including adaptation to local environment and production conditions, market demand, culinary and cultural needs, livestock feed, construction and soil fertilization. [2] Physical existence of desired seeds in enough quantity in a reasonable proximity (spatial availability) for critical sowing periods (temporal availability). [3] Means to acquire seeds such as cash, credit, social network and transportation with affordability and awareness/information. [4] Seed is healthy (free from disease/pest), has good physical qualities (not broken/cracked/shriveled), has good genetic and physiological qualities (good germination, optimum moisture content, genetic purity and vigor), free from weeds and poses preferred color/size/shape/taste.

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
