# Peer review of "Pluralistic Seed System Development: A Path to Seed Security?"

_agronomy, doi:10.3390/agronomy11020372_

Round 1

Reviewer 1 Report

This article uses a solid comparative design (with two contrasting districts) to explore and analyse the seed systems used for teff and wheat.  This is solid, adapting field methods from Seed System Security Assessments and similar tools, to paint a picture of the seed security issues faced by different types of farmers.  While the analysis of social disaggregation arguably could have been taken further through the use of regressions or similar inferential methods, the way it was presented here made the authors' broad point - that social difference matters in seed security. The analysis of farmers' seed security was combined with an assessment of the functioning of the various seed systems (formal, informal, intermediate), based on Key Informant discussions, and a discussion of how well the Pluralistic Seed Sector Development Strategy addressed constraints.  This was very well done, and allowed paper to make cogent and valid points about gaps in policy implementation, and to pinpoint elements that need further development.  This is handled well, and makes a useful contribution to the literature. 

Some general comments:

  • when discussing policy constraints, and highlighting some of the challenges for better governance (and policy reform) in the specific setting of Ethiopia, it would help to at least point to some literature that explores the nature of the state in Ethiopia, in terms of how it manages seed systems (e.g. Hassena Beko's 2017 PhD https://library.wur.nl/WebQuery/wurpubs/526078, or in relation to the input package programme, e.g. Keeley, James, and Ian Scoones. "Knowledge, power and politics: the environmental policy-making process in Ethiopia." Journal of Modern African Studies (2000): 89-120."). While I do not think you need to engage in depth with the analysis of policy processes  or legal pluralism in this literature, at least taking note of it could help highlight the need for understanding and contending with specific national contexts to have policy reform that is effective.  For example, in line 751, the paper suggests that greater awareness of PSSDS is enough to overcome coordination problems between different NGO and government actors. However, in light of the discussion of policy and governance in the discussion later on, and the acknowledgement that these groups have different goals, simply referring to PSSDS will not be enough. 
  • The different seed systems are treated separately in the discussion. Do any of the policy responses under PSSDS span more than one system? Could recommendations be made on this? 
  • Methods can be explained slightly better.  185-197:  this is quite condensed, and without reference to the RTB source, the themes of the questioning is unclear: these key informants are reporting on what?  Is it in reference to a specific time period (as with farmers?), or specific crops?  Additionally, the agro-ecologies of the two districts were not clear - some further information on average land holdings, soil types and quality, elevation, etc. would be useful.  The maps are good, but the legend is hard to read and the symbols too similar. 
  • Discussion around gaps in breeding. More crop development is generally needed in most settings, but teff and wheat may be somewhat special cases, because the former is endemic and under-researched until recently, and the latter is facing serious issues with rust. For other crops (maize, sorghum, potatoes, pulses, oilseeds) there may be more released varieties on the shelf that are potentially useful, placing the challenge more on variety promotion, seed production and dissemination.  (see, for example, Ojiewo, C. O., et al. Mainstreaming efficient legume seed systems in Eastern Africa: challenges, opportunities and contributions towards improved livelihoods. FAO, 2018.). My point here is that the focus on breeding gaps may be a) crop-specific and b) with the centralised and top-down approach to Research and Extension in Ethiopia, there may be released varieties (even for teff) that would be popular in some districts - if farmers could only see them via demonstration plots or other outreach strategies.

Minor points

  • 12 - not clear what PSSDS is
  • 46 - approve varieties, not seeds
  • 61-62 FAO did not 'add' two dimensions: varietal suitability had previously been there, under 'genetic quality'
  • 97 - note that the 'normal conditions' of these regions are not that different from some of the areas of SSSAs or SSAs - e.g. assessments in Kenya, Tanzania, Malawi, Zambia, Zimbabwe, for example.  These seed security assessments in stressed areas conclude that a (and often the) main issue is being under-served by formal and intermediate seed systems. 
  • Little in-depth study? Ethiopian seed systems are more studied than  most in Africa, due to links with ISSD, and its status as a centre of crop genetic diversity.
  • 165 - QDS could be explained a bit more
  • 180 - what sorts of interventions?
  • 254 - clarify if seed lot or variety.  For example, a variety is still 'improved' if it has been recycled 4 seasons, though the seed system supplying it is now 'informal'? 
  • 287 - how was triangulation done. How was 'preferred' determined in practice? 
  • 323-331 - striking that sorghum was abandoned, as there are many local varieties. This shift may be more likely due to other drivers of change, such as rates of return to that crop, or labour demands for sorghum production   
  • 442 Heexosa - interesting that they highlighted the limits of CSBs and an emergency seed reserve, but not the other district.  Why might that be? This critique could be brought forward to the PSSDS's plans for a national emergency seed reserve (724), which would face similar constraints of quality, value for money, and effectiveness. 
  • 455 0 Access is not just an emergency issue - weak extension and seed dissemination affect access in other contexts
  • Table 3 - how untangle access from demand for a specific variety?  (see 518-19: it could be that poorer HHs prefer a lower value variety for consumption, rather than a higher value one for sales?)
  • 490 price info is good. While grain prices may be higher if the variety is a desired one (price variations in local markets by variety are common - see Sperling et al 2020, for a wheat trader <100km away from Hexoosa), the paper's account shows for availability of new varieties (and constraints to supply through outgrowers) can drive up prices. 
  • Gender - good qualitative information and interesting arguments. 
  • 654 - mixing - may say more about the differences in harvest practices by crop. Is teff still harvested by hand in Hexoosa? 
  • 741 - should be 'riven'
  • 744 - motivations of DAs. The literature on the Sasakawa/Global 2000 package programme and extension echoes this as well, and highlights the incentives for field  agents to meet quotas
  •  

.

Author Response

Response to Reviewers’ Comments

I am writing on behalf of the authors to submit our revised manuscript entitled, “Pluralistic seed system development: a path to seed security?,” which has been accepted with minor revisions for publication in agronomy.

We are very thankful to both reviewers for their high quality and clearly written comments. We see the comments are coming from reviewers who knows both the literature and the country context. We believe that their inputs helped us to further improve the manuscript. In the revised version, we have provided clarifications, descriptions, and additional information in line with the recommendation and references provided by the reviewers.

We have outlined our response to both reviewers’ comments as shown below. Line number in some of our responses refer to the line number in the revised version of the manuscript.

Point 1: This article uses a solid comparative design (with two contrasting districts) to explore and analyse the seed systems used for teff and wheat.  This is solid, adapting field methods from Seed System Security Assessments and similar tools, to paint a picture of the seed security issues faced by different types of farmers.  While the analysis of social disaggregation arguably could have been taken further through the use of regressions or similar inferential methods, the way it was presented here made the authors' broad point - that social difference matters in seed security. The analysis of farmers' seed security was combined with an assessment of the functioning of the various seed systems (formal, informal, intermediate), based on Key Informant discussions, and a discussion of how well the Pluralistic Seed Sector Development Strategy addressed constraints.  This was very well done and allowed paper to make cogent and valid points about gaps in policy implementation, and to pinpoint elements that need further development.  This is handled well, and makes a useful contribution to the literature. 

Response 1: We are glad to hear this.

Point 2: When discussing policy constraints, and highlighting some of the challenges for better governance (and policy reform) in the specific setting of Ethiopia, it would help to at least point to some literature that explores the nature of the state in Ethiopia, in terms of how it manages seed systems (e.g. Hassena Beko's 2017 PhD https://library.wur.nl/WebQuery/wurpubs/526078, or in relation to the input package programme, e.g. Keeley, James, and Ian Scoones. "Knowledge, power and politics: the environmental policy-making process in Ethiopia." Journal of Modern African Studies (2000): 89-120."). While I do not think you need to engage in depth with the analysis of policy processes  or legal pluralism in this literature, at least taking note of it could help highlight the need for understanding and contending with specific national contexts to have policy reform that is effective.  For example, in line 751, the paper suggests that greater awareness of PSSDS is enough to overcome coordination problems between different NGO and government actors. However, in light of the discussion of policy and governance in the discussion later on, and the acknowledgement that these groups have different goals, simply referring to PSSDS will not be enough.

Response 2: This is an important point and we have addressed it in the revised version. We added a brief explanation of why implementation of PSSDS is lagging referencing some of the suggested literature. – Line 801

Point 3: The different seed systems are treated separately in the discussion. Do any of the policy responses under PSSDS span more than one system? Could recommendations be made on this?

Response 3: In our findings, we show how farmers span more than one system when they use seed from the different systems. The policy responses under PSSDS span more than one system. We added PSSDS interventions for cross-linkage under each seed system and highlighted in our discussions. – Line 756, 872, 880, 1041, 1094

Point 4: Methods can be explained slightly better.  185-197:  this is quite condensed, and without reference to the RTB source, the themes of the questioning is unclear: these key informants are reporting on what?  Is it in reference to a specific time period (as with farmers?), or specific crops?  Additionally, the agro-ecologies of the two districts were not clear - some further information on average land holdings, soil types and quality, elevation, etc. would be useful.  The maps are good, but the legend is hard to read and the symbols too similar.

Response 4: We revised the methods section, including more details about the use of the RTB framework and the topics covered in the FGDs and key informant interviews.  We have also added a table in the Study Area section presenting information on the demographic and agroecological characteristics of the two districts. We changed the font size of the map legends for better readability. Line – 193, 262

Point 5: Discussion around gaps in breeding. More crop development is generally needed in most settings, but teff and wheat may be somewhat special cases, because the former is endemic and under-researched until recently, and the latter is facing serious issues with rust. For other crops (maize, sorghum, potatoes, pulses, oilseeds) there may be more released varieties on the shelf that are potentially useful, placing the challenge more on variety promotion, seed production and dissemination.  (see, for example, Ojiewo, C. O., et al. Mainstreaming efficient legume seed systems in Eastern Africa: challenges, opportunities, and contributions towards improved livelihoods. FAO, 2018.). My point here is that the focus on breeding gaps may be a) crop-specific and b) with the centralized and top-down approach to Research and Extension in Ethiopia, there may be released varieties (even for teff) that would be popular in some districts - if farmers could only see them via demonstration plots or other outreach strategies.

Response 5: In this paper we do highlight some constraints related to variety deployment, especially for wheat. We have added a sentence indicating that variety deployment is an important issue for many other crops. However, we feel there is not scope within this article for a more in-depth discussion. Line – 928

Point 6: 12 - not clear what PSSDS is

Response 6: We feel there is not scope in the abstract for a detailed explanation of the PSSDS. However, the explanation comes in the first paragraph of the introduction. Line – 36

Point 7: 46 - approve varieties, not seeds

Response 7: corrected. -  Line 47 

Point 8: 61-62 FAO did not 'add' two dimensions: varietal suitability had previously been there, under 'genetic quality'

Response 8: corrected. Line - 62

Point 9: 97 - note that the 'normal conditions' of these regions are not that different from some of the areas of SSSAs or SSAs - e.g. assessments in Kenya, Tanzania, Malawi, Zambia, Zimbabwe, for example.  These seed security assessments in stressed areas conclude that a (and often the) main issue is being under-served by formal and intermediate seed systems

Response 9: Noted the reviewer’s comment and agree with the reference made to the assessment results in other African countries. However, we used the expression to characterize the study districts and the study period that it is not non-disaster/post-disaster situation - Line 100

Point 10: Little in-depth study? Ethiopian seed systems are more studied than most in Africa, due to links with ISSD, and its status as a centre of crop genetic diversity

Response 10: We agree with the reviewer’s comment that Ethiopia’s seed systems are well studied. But our point was that the seed security framework has rarely been used to analyse Ethiopia’s seed systems. We have clarified this in the text. Line - 100

Point 11: 165 - QDS could be explained a bit more

Response 11: QDS explained more in detail as suggested by the reviewer. -  Line 167

Point 12: 180 - what sorts of interventions?

Response 12: The interventions are discussed in detail in section 6, as the following sentence in the manuscript indicates. We feel it is not necessary to mention them here. -  Line starting 728 

Point 13: 254 - clarify if seed lot or variety.  For example, a variety is still 'improved' if it has been recycled 4 seasons, though the seed system supplying it is now 'informal'?

Response 13: We clarified the wording in the footnote #5 on page 8

Point 14: 287 - how was triangulation done. How was 'preferred' determined in practice?

Response 14: While varietal preference was determined based on the proportion of farmers who gave high score for a variety based on the pre-defined criteria. This was cross-checked against the FGD qualitative ranking. We did not find any inconsistencies so in practice preference was determined using the HH survey ranking. We added a brief clarification in the text. Line 319

Point 15: 323-331 - striking that sorghum was abandoned, as there are many local varieties. This shift may be more likely due to other drivers of change, such as rates of return to that crop, or labour demands for sorghum production  

Response 15: Yes, this was a striking information that farmers provided. Farmers explained more on this, but we had to cut the following quote to keep the focus on lack of improved varieties.  “Especially we loved cultivating sorghum because we can make several foods and drinks from its grain. Sorghum has a good yield and its cultivation is not labour intensive. Farmers who did not have oxen for ploughing and enough labour could easily grow sorghum. In our area, the poor ate sorghum and maize, and they sold tef for income. We call sorghum a poor man’s friend. As the fertility of our soil declined, new plant diseases emerged (e.g. wheat rust and striga) and climatic conditions varied (e.g. late onset of rain), many local varieties of our crops failed to adapt. Sometimes we experienced total crop loss.”  In addition, sorghum diversity in Gindabarat is not as high as north central, western, and eastern part of Ethiopia.

Point 16: 442 Heexosa - interesting that they highlighted the limits of CSBs and an emergency seed reserve, but not the other district.  Why might that be? This critique could be brought forward to the PSSDS's plans for a national emergency seed reserve (724), which would face similar constraints of quality, value for money, and effectiveness.

Response 16: Farmers in Heexosa did not mention limits of CSBs here, they only referred to the seed reserve which is less adaptable due to short wheat storability. We have added a sentence about this in the discussion on PSSDS’s plans for a national emergency seed reserve. Line - 835

Point 17: 455 Access is not just an emergency issue - weak extension and seed dissemination affect access in other contexts

Response 17: Yes, this was clearly articulated in the CGIAR publication we cited. We added a brief clarification. Line 478 

Point 18: Table 3 (revised and NOW TABLE 4) - how untangle access from demand for a specific variety?  (see 518-19: it could be that poorer HHs prefer a lower value variety for consumption, rather than a higher value one for sales?)

Response 18: Yes, the poorer HHs preferred Daaboo that has lower market value for consumption, rather than Quncho that has a higher market value. We explained the same information in terms of teff grain price as input, in which case the high price of Quncho limited access for poor farmers.

Point 19: 490 price info is good. While grain prices may be higher if the variety is a desired one (price variations in local markets by variety are common - see Sperling et al 2020, for a wheat trader <100km away from Heexosa), the paper's account shows for availability of new varieties (and constraints to supply through outgrowers) can drive up prices

Response 19: We agree with the comments made. We feel that the first point regarding high price of desired variety is addressed through our discussion of Quncho variety and increase in price linked to constraints to supply throughout-growers.

Point 20: Gender - good qualitative information and interesting arguments.

Response 20: Yes, it was unexpected and interesting information that we found

Point 21: 654 - mixing - may say more about the differences in harvest practices by crop. Is teff still harvested by hand in Heexosa?

Response 21: Yes, teff is still harvested by hand in both districts. We added a brief clarification in the text. Line - 680 

Point 22: 741 - should be 'riven'

Response 22: corrected. -  Line 772 

Point 23: 744 - motivations of DAs. The literature on the Sasakawa/Global 2000 package programme and extension echoes this as well, and highlights the incentives for field agents to meet quotas

Response 23: We referred to the recommended literature. Line - 777

Reviewer 2 Report

The article is well documented and written, presenting this innovative approach of pluralistic seed systems. This approach is well grounded in the analysis done by the authors in the two areas in Ethiopia. Even the new Ethiopian seed policies are well presented. 

The only point I would add to the paper, maybe in the conclusions, is about the role of donors to seed system development. The Authors analyzed the role of NGOs but it is missing some notes about the role of external private and public donors on Ethiopian seed systems and if they are following the new Ethiopian seed policies. To my knowledge in some regions the role of donors (e.g. private foundations or national governments aid programs) is more influencing than internal supporting measures. Can you add a paragraph on that point under the paragraph 7 Conclusions? What do you think about the role of these donors? 

Author Response

Response to Reviewer 2 Comments

Point 1: The article is well documented and written, presenting this innovative approach of pluralistic seed systems. This approach is well grounded in the analysis done by the authors in the two areas in Ethiopia. Even the new Ethiopian seed policies are well presented.

Response 1: We are glad to hear this.

Point 2: The only point I would add to the paper, maybe in the conclusions, is about the role of donors to seed system development. The Authors analysed the role of NGOs, but it is missing some notes about the role of external private and public donors on Ethiopian seed systems and if they are following the new Ethiopian seed policies. To my knowledge in some regions the role of donors (e.g. private foundations or national governments aid programs) is more influencing than internal supporting measures. Can you add a paragraph on that point under the paragraph 7 Conclusions? What do you think about the role of these donors?

Response 2: The point about the donors is well taken. We feel that this role is highlighted in terms of actors’ activities at local level in Appendix A, Table A1. These are NGOs supporting community-based seed systems (CSBs and SPCs). We have also highlighted how the long-term donor support in Heexosa that seem to have led to gender empowerment. We have also highlighted the role of ISSD program in shaping the PSSDS itself. However, we did not feel to include the political economy of seed for this paper. In the revised version, we have added a sentence wording on this issue for the purpose of this article also in the main text.
